# Ultracompact mirror device for forming 20-nm achromatic soft-X-ray focus toward multimodal and multicolor nanoanalyses

Takenori Shimamura [1,2,3] ✉, Yoko Takeo [2,3], Fumika Moriya [1], Takashi Kimura [3], Mari Shimura [4,5], Yasunori Senba [2,4], Hikaru Kishimoto [2], Haruhiko Ohashi [2,4], Kenta Shimba [1], Yasuhiko Jimbo [1] & Hidekazu Mimura [4,6] ✉

Nanoscale soft-X-ray microscopy is a powerful analysis tool in biological, chemical, and physical sciences. To enhance its probe sensitivity and leverage multimodal soft-X-ray microscopy, precise achromatic focusing devices, which are challenging to fabricate, are essential. Here, we develop an ultra-compact Kirkpatrick-Baez (ucKB) mirror, which is ideal for the high-performance nanofocusing of broadband-energy X-rays. We apply our advanced fabrication techniques and short-focal-length strategy to realize diffraction-limited focusing over the entire soft-X-ray range. We achieve a focus size of 20.4 nm at 2 keV, which represents a significant improvement in achromatic soft-X-ray focusing. The ucKB mirror extends soft-X-ray fluorescence microscopy by producing a bicolor nanoprobe with a 1- or 2-keV photon energy. We propose a subcellular chemical mapping method that allows a comprehensive analysis of specimen morphology and the distribution of light elements and metal elements. ucKB mirrors will improve soft-X-ray nanoanalyses by facilitating photon-hungry, multimodal, and polychromatic methods, even with table-top X-ray sources.

Chromatic aberration, which appears as color fringing due to a shift of focus with wavelength, is an intrinsic property of light focusing. Such stretched foci impair high-resolution imaging using polychromatic or broadband light sources. Visible-light microscopes and telescopes overcome this physics barrier by employing achromatic doublet lenses or Newtonian and Cassegrain reflectors, which date back to the 17th and 18th centuries[1,2].

Bright monochromatic X-rays in the soft-X-ray region (low-energy range of 0.3 to 2 keV) have been generated using synchrotron radiation (SR), allowing soft-X-ray microscopy to use diffractive focusing devices. Despite their inherent chromatic aberration, such devices can focus SR-based monochromatic soft X-rays to a sub-10-nm spot size[3]. However, chromatic aberration causes a dependence of the focus position on the X-ray wavelength (photon energy). Energy scanning thus requires samples to be repositioned throughout or between the spectroscopy procedures. Chromatic aberration also imposes mono-chromaticity on photon-hungry methods that can use polychromatic X-rays. Soft-X-ray fluorescence microscopy, for instance, increases the fluorescence count by expanding the irradiated sample area to the sub-micrometer level[4,5] or extending the measurement time[6] rather than by enhancing the photon flux using polychromatic or broadband energy nanoprobes. These drawbacks of diffractive focusing devices hinder

[1]School of Engineering, The University of Tokyo, 7-3-1 Hongo, Bunkyo, Tokyo 113-8656, Japan. [2]Japan Synchrotron Radiation Research Institute, 1-1-1 Koto, Sayo, Sayo District, Hyogo 679-5198, Japan. [3]The Institute for Solid State Physics, The University of Tokyo, 5-1-5 Kashiwanoha, Kashiwa, Chiba 277-8581, Japan. [4]RIKEN SPring-8 Center, 1-1-1 Koto, Sayo, Sayo District, Hyogo 679-5148, Japan. [5]Department of Refractory Viral Infection, Research Institute, National Center for Global Health and Medicine, 1-21-1 Toyama, Shinjuku, Tokyo 162-8655, Japan. [6]Research Center for Advanced Science and Technology, The University of Tokyo, 4-6-1 Komaba, Meguro, Tokyo 153-8904, Japan. ✉e-mail: tshimamura@issp.u-tokyo.ac.jp; mimura@upm.rcast.u-tokyo.ac.jp

multimodal and multi-energy soft-X-ray nanoanalyses even though the structural[7], chemical[8], elemental[9], and magnetic[10] composition of samples can be determined in the soft-X-ray region. As is the case for the visible-light regime, achromatic focusing devices are essential for advancing nanoscale soft-X-ray microscopy beyond the limit of chromatic aberration.

As achromatic lenses significantly absorb soft X-rays[11], solely reflective devices are practical for achromatic soft-X-ray nanoprobes. Soft-X-ray nanofocusing requires a ten to twenty times larger numerical aperture (NA) than that for nanofocusing in the hard-X-ray region (high-energy range of >10 keV). A large NA necessitates increasing the grazing angle of X-ray mirrors, for instance from 1 to 25 mrad, which makes the focusing wavefront susceptible to surface figure errors (see Methods for $\theta_0$). Nanoscale fabrication technologies have been developed to smooth traditional hard-X-ray focusing mirrors[12,13] and to precisely form large-NA X-ray mirrors[14–16]. Nevertheless, rigorous fabrication requirements have prevented advanced X-ray mirrors from achieving the ideal soft-X-ray focus size[15–20]. Achromatic soft-X-ray nanofocusing is barely diffraction-limited with a focus size of 241 nm × 81 nm at 0.3-keV photon energy[18]. The focus size does not shrink, but instead expands, with shortening X-ray wavelength[19,20] because both the diffraction limit and the wavefront error tolerance for the diffraction limit are proportional to the X-ray wavelength[1]. Currently available achromatic soft-X-ray nanoprobes are severely limited by the fabrication process used.

To realize ideal nanofocusing over the entire soft-X-ray range, novel strategies have to be adopted in addition to conventional straightforward development. Due to the grazing angle limitation in the hard-X-ray region, X-ray mirrors have been extended to 1 m for large spatial acceptance while maintaining nanoprecision[21,22]. With the large grazing angle allowed in the soft-X-ray region, millimeter-scale mirrors moderately accept X-rays and achieve focusing throughput superior or comparable to that for diffractive focusing devices[3,8]. Such mirrors can be precisely fabricated because only the middle- to high-frequency range (1 cm$^{-1}$ to 10 $\mu$m$^{-1}$) is crucial for their smooth and freeform shapes (low-frequency figure errors on millimeter-scale mirrors are considered to be linear offsets). X-ray mirrors have been designed with long focal lengths (>100 mm) for diverse detector-sample arrangements[23]. In contrast, short mirrors can bring their foci much closer to the mirror center. Millimeter-scale focal lengths enhance focusing robustness as X-rays reach the focal plane before widely spreading due to mirror figure errors (see Methods).

Such ultracompact mirrors can be simple alternatives for ideal nanofocusing. Nevertheless, their fabrication is even more challenging than that of conventional mirrors, as short focal lengths proportionally decrease the tangential radii of curvature (ROCs), making existing fabrication techniques less effective. A robust focusing strategy, reinforced by new technology, should enable the realization of ideal achromatic soft-X-ray nanoprobes.

Here, besides developing fabrication techniques for highly curved mirrors, we examine remarkably compact mirrors and short focal lengths for nanofocusing soft X-rays. The proposed mirror system employs a sequentially crossed arrangement referred to as the Kirkpatrick-Baez (KB) geometry[24] (see below), which simplifies the shape of the individual mirrors. An ultra-compact KB (ucKB) mirror with component focal lengths of 2 and 8 mm is designed based on a short-focal-length strategy and then fabricated using our advanced methods. The ucKB mirror achieves a sub-50-nm nanoprobe with a photon energy of more than 1 keV, which gives access to mesoscopic scales below 100 nm, where magnetic skyrmions[10] and subcellular biological behavior[25] between bulk and nanoscale properties emerge. We apply the ucKB mirror to enhance soft-X-ray fluorescence microscopy, namely the low-energy X-ray fluorescence (LEXRF) technique[26]. The results of observations of fixed biological specimens demonstrate the feasibility of a multimodal and multi-color soft-X-ray nanoanalysis.

## Results

### Design of ultracompact X-ray mirrors

Figure errors cause an uneven slope distribution, which scatters rays in proportion to the focal length. From a geometrical viewpoint, short focal lengths mitigate the effects of mirror defects on ray deflection (see Methods). The focus spot diffracted by figure errors is deduced using wave optics:

$$\Delta u \approx \frac{\lambda r}{d_m \theta_0} , \qquad (1)$$

where $\Delta u$ is the gap between the diffracted focus spots, $\lambda$ is the wavelength of the monochromatic X-rays, $r$ is the distance between the focus and the downstream mirror end, $d_m$ is the spatial wavelength of a periodic figure error, and $\theta_0$ is the grazing angle of X-ray mirrors (see Methods). In principle, foci close to the devices are beneficial for

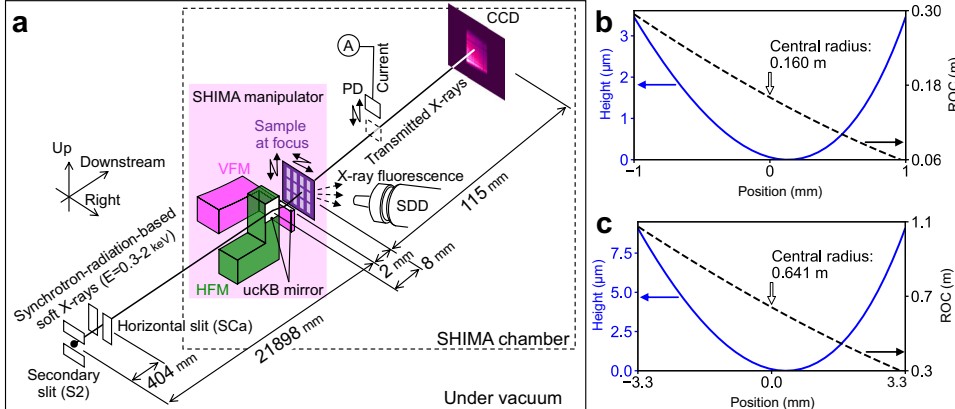

**Fig. 1 | Experimental configuration using ultracompact Kirkpatrick-Baez (ucKB) mirror composed of vertically focusing mirror (VFM) and horizontally focusing mirror (HFM). a** Layout of soft-X-ray microscope. The transmitted X-rays and diffraction patterns are acquired with a charge-coupled device (CCD) X-ray camera. Low-energy X-ray fluorescence microscopy simultaneously utilizes a silicon drift detector (SDD) and a photodiode (PD) to capture X-ray fluorescence and absorption, respectively. Designed elliptic figure profiles and sub-meter radii of curvature (ROCs) of (**b**) VFM and (**c**) HFM. With the positional origin at the mirror center, the positive *x*-direction is toward the downstream side of the mirror.

**Table 1 | Design parameters for ultracompact X-ray mirrors**

| Parameter | | Units | VFM | HFM |
|---|---|---|---|---|
| Reflective area | Surface profile | | Elliptic cylinder | |
| | Surface coating | | Ni (purity: 4N) | |
| | Grazing angle[a] | mrad | 25 | |
| | Effective mirror length | mm | 2 | 6.6 |
| | Focal length (f) | mm | 2 | 8 |
| | Working distance (r) | mm | 1 | 4 |
| | Incident flight path[b] | mm | 21898 | 21488 |
| | Semimajor axis | mm | 10950 | 10748 |
| | Semiminor axis | mm | 5.2313 | 10.364 |
| | Tangential ROC[a] | m | 0.160 | 0.641 |
| Substrate | Material | | Synthetic fused silica | |
| | Shape | | L-shaped cylinder (mirror and ballast parts) | |
| | Tangential ROC | m | 0.160 | 0.620 |
| | Mirror-part dimensions (L × W × T) | mm³ | 2.5 × 20 × 10 | 8.5 × 20 × 10 |
| Focusing performance | Reflectivity[a] | % | 50.9[c], 60.8[d], 56.3[e] | |
| | Spatial beam acceptance | μm | 51.6 | 168.2 |
| | Geometrical magnification | | 10949 | 2686 |
| | Numerical aperture | | 0.0149 | 0.0116 |
| | Focus size[c,f] | nm | 36.9 | 48.8 |
| | Depth of focus (Rayleigh range)[c,f] | μm | ± 4.4 | ± 7.2 |

[a]Value at mirror center.
[b]See Methods for underlying design.
[c]Value at 1-keV photon energy.
[d]Value at 1.5-keV photon energy.
[e]Value at 2-keV photon energy.
[f]Value is inversely proportional to photon energy.

realistic nanofocusing systems. Equation (1) with empirical values shows that soft-X-ray nanofocusing mirrors for mesoscopic scales require $r = 1$ mm (see Methods). Diffractive nanofocusing devices often employ an $r$ value of tens to hundreds of micrometers[3]. However, an $r$ value of 1 mm goes against the traditional design philosophy for hard-X-ray mirrors.

As illustrated in Fig. 1a, the KB geometry simplifies a doubly curved focusing mirror to a pair of elliptic-cylindrical surfaces, namely a vertically focusing mirror (VFM) and a horizontally focusing mirror (HFM). However, one mirror comes between the other mirror and the focus, and the downstream mirror prevents a short upstream focal length. To achieve an extremely short focal length for both mirrors of the ucKB mirror, the downstream mirror length was reduced to 2 mm. Table 1 lists the detailed optical design parameters for the VFM and HFM. The $r$ value for the VFM is 1 mm. The ucKB mirror was designed to concentrate 1-keV soft X-rays into a sub-50-nm focus spot, which is sufficient for mesoscopic-scale analyses.

The large NA and demagnification factor require a sub-meter-radius design, as shown in Fig. 1b and c. Such strongly curved mirrors are the most critical challenge in terms of advanced production and metrology.

**Mirror fabrication**
We developed a figure correction method with a sub-millimeter spatial resolution and simple interferometry, both of which were specifically designed for millimeter-long sub-meter-radius mirrors. This combination overcomes the technical obstacles, allowing the production of the first pair of sub-meter-radius mirrors, as shown in Fig. 2a.

Angstrom-scale smooth cylindrical surfaces were efficiently transformed into the designed elliptic-cylindrical reflective surfaces through Ni deposition. Figures 2b and c show that the residual figure errors were controlled to within 0.5% of the maximum Ni film thickness, which can reach several micrometers. Compared to the traditional differential deposition technique[27], the present figure correction achieves a higher aspect ratio (length to height) for the VFM film thickness profile (50000:1 vs. 1000:1). Rayleigh's quarter wavelength rule is an empirical criterion for an ideal wavefront at the focus[1]. Considering this rule and the peak-to-valley (PV) residual errors (2.8 and 11.8 nm for the VFM and HFM, respectively), the VFM can ideally focus soft X-rays below 2-keV photon energy. See Methods for details of the developed production and measurement process.

**Achromatic sub-50-nm soft-X-ray probes**
The ucKB mirror was evaluated at beamline BL25SU-A, SPring-8, Japan, with a photon-energy tunability from 0.3 to 2 keV[28]. The ucKB mirror was mounted in the soft-X-ray high-resolution mapping (SHIMA) system, which is detailed in Methods.

As shown in Fig. 3a, the focus position formed by the ucKB mirror stays mostly constant over different photon energies. The nanobeam shrinks with increasing photon energy. The difference in focus size between ptychography and knife-edge scanning is due to the blunt knife edge. Nevertheless, knife-edge scanning at 1-keV photon energy has a focus size of 77.4 nm × 96.7 nm in terms of the full width at half maximum (FWHM). The large figure error for the HFM causes the horizontal focus size to not reach the design value at 2-keV photon energy. The ptychographic focus size generally falls in the design value range. As shown in Fig. 3b, the ptychographically reconstructed profiles exhibit subtle sidelobes at 2-keV photon energy; however, the nanobeam is focused to a spot size of 20.4 nm × 40.7 nm. The large NA for the ucKB mirror allows its vertical focus size to reach the minimum size that total-reflection mirrors can practically attain in the hard-X-ray region (25 nm)[29]. The ratio of the flux within the rectangular FWHM focus area was over 40% at a photon energy of 1 keV or less. See Methods and Supplementary Section 3 for details of the focusing performance.

To verify the formation of this achromatic nanoprobe, Au nanoparticles were observed, as shown in Fig. 3c. After the ptychography image in Fig. 3d was used to locate four condensed particles at 1-keV photon energy, scanning transmission X-ray microscopy (STXM) was performed at 2 keV with a 15-nm scanning pitch. Figure 3e compares the particle observed by scanning electron microscopy (SEM) and STXM. The elongated X-ray nanoprobe results in a lower horizontal resolution in the STXM image. The right sidelobe in the horizontal focusing profile in Fig. 3b grazed the particle, resulting in satellite signals on the left-hand side of the image (see arrows). Nevertheless, the boundary of the particle is sharp vertically; the STXM image shows even small bump outlines at the top and bottom of the particle.

The figure errors for the VFM and HFM were determined based on the ptychography results, as shown in Fig. 3f. The interferometry and ptychography results agree within 3 nm, showing that the wavefields were successfully reconstructed. The root-mean-square surface roughness in a 500-nm square area of the mirror was 0.43 nm, which is small enough to prevent X-ray scattering. It is thus concluded that the focus size was precisely evaluated using ptychography and that the ucKB mirror can concentrate X-rays into sub-50-nm spots. See Methods and Supplementary Section 2 for details of the preparation of test specimens and reconstructed figure errors, respectively.

**Soft-X-ray fluorescence microscopy down to 100-nm resolution**
As the focus position is constant, the ucKB mirror can achromatically nanofocus a series of soft X-rays whose photon energy varies even by as much as a factor of two, as shown in Fig. 4a. Such multicolor nanoprobes enhance the LEXRF technique[26], which suffers from low

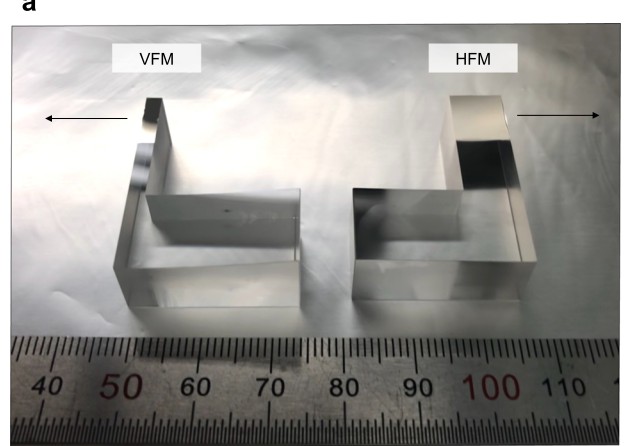
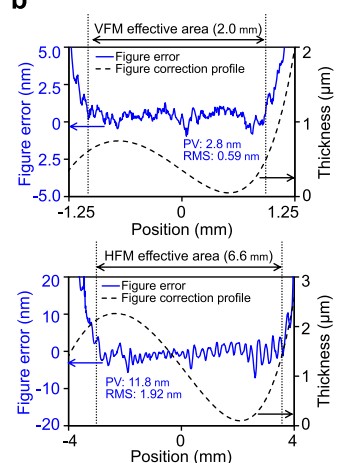

**Fig. 2 | Fabricated mirror pair for ultracompact Kirkpatrick-Baez mirror.**
**a** Elliptic cylindrical reflective surfaces fabricated on L-shaped glass cylinder substrates (ruler units: mm). Arrows indicate X-ray path. **b** Film thickness profile for figure correction of substrates (dotted line) and residual figure errors (solid line). The effective area reflected soft X-rays in the experiments. Figure errors within this area are given in terms of peak-to-valley (PV) and root-mean-square (RMS) values. With the positional origin at the substrate center, the positive $x$-direction is toward the downstream side of the mirror.

fluorescence yields for light elements[30]. Higher-energy X-rays can stimulate fluorescence from a wider range of chemical elements. However, fluorescence emission events for light elements such as C, N, and O are typically 5 and 30 times more sporadic at 1 and 2 keV, respectively, than those for metals. With the incident photon energy tuned to 1 keV, a sub-100-nm bicolor nanoprobe composed of 1-keV fundamental X-rays and 2-keV harmonics was produced using the ucKB mirror to stimulate fluorescence from both light elements and metal elements. To demonstrate the effectiveness of the LEXRF technique with a bicolor nanoprobe, Hep G2 cells and primary hippocampal neurons were chemically fixed and specimens were prepared, as shown in Fig. 4b and c, respectively.

Correlating the element distribution with morphology requires the sample thickness to be known. The sample thickness should be independently obtained using a technique such as atomic force microscopy[31] because the X-ray absorption results represent the product of the mass absorption coefficient, density, and thickness. The iterative algorithm proposed in the present study (see Methods) can evaluate these parameters separately using solely X-ray-based results.

Figure 5a–d show X-ray absorption and fluorescence images of Hep G2 cells. The distributions of C, N, and O, which constitute proteins, agree with the overall morphology. Zn is spread over the cells and concentrated around the nuclei. The 1.739-keV fluorescence from Si increases at the window frame, demonstrating the successful production of a bicolor X-ray probe. Figure 5d shows biologically produced granules within a field of view (FOV) of 5 μm. Without sectioning the sample, our LEXRF technique revealed that the granules were hollow and surrounded by light elements. Such observations are possible only with nanoscale LEXRF microscopy.

The sample thickness was calculated using mass information and X-ray absorption. For the area with more than 1% X-ray transmittance in Fig. 5a, the angle between the Si window frame and the supporting film is in good agreement with the theoretical value (54.0 degrees vs. 54.7 degrees, see Fig. 4a). The noise level for X-ray absorption is lower than that for fluorescence (0.1% vs. 10%). The precisely calculated thickness can be combined with the mass thickness to obtain concentration information, as shown in Fig. 5e–h. The oxygen is concentrated in the vicinity of the nucleus in Fig. 5e and f. In Fig. 5h, the O concentration and is more homogeneous than the C and N concentrations, which increase in areas with a small sample thickness.

Figure 6a–c show the mass thickness for the neurons. Iron can be observed in part of the soma in Fig. 6a. In Fig. 6d, the thickness of the

neuron is slightly lower than that estimated using visible-light microscopy (3.4 μm vs. 4.0 μm). As the X-ray intensity reflects the net thickness, the difference may have resulted from the chemical fixation and air-drying processes, which can leave sparse areas inside the cell. Regarding the neurites in Fig. 6b, c, spine-like protrusions exist around the swelling, which is called a varicosity (see arrows). These structures can be involved in synaptic activities linked to Cu and Fe[32–34], which are localized in Fig. 6b. Varicosity emergence correlates with nearby spine loss[35]. Copper signals near the varicosity were thus evaluated at 100-nm resolution, as shown in Fig. 6c and f. Zinc is homogeneously distributed, in contrast to Cu and Fe. The concentration analysis shows that the bump area contains more Cu and Fe than can be explained by the thickness variation.

## Discussion

The ucKB mirror was developed by leveraging a short-focal-length strategy and our advanced fabrication method for sub-meter-radius surfaces. The nanobeam was not fabrication-constrained but nearly diffraction-limited over the entire soft-X-ray range. A sub-50-nm achromatic focus size was attained above 1-keV photon energy. The defocus of the achromatic nanobeam was within the Rayleigh range. Considering that the beam size expands in proportion to NA and defocus distance, the ucKB mirror is a strong candidate for polychromatic soft-X-ray nanofocusing. Whereas the present study adopted a multi-energy or multimodal analysis using STXM, ptychography, and LEXRF, the achromatic nanoprobe allows X-ray absorption spectroscopy to be readily performed without adjusting the focus position following other X-ray microscopy observations, for instance to further examine the chemical states of a single aerosol nanoparticle[36].

X-rays uniquely allow nanoscale chemical mapping within unsectioned and unlabeled biological samples. The Zn distribution in Fig. 5a–c and the metal locations in Fig. 6a–c are consistent with previous studies that used conventional hard-X-ray fluorescence (XRF) microscopy[37–40]. This nanoscale method can only map the amount of high-Z elements; X-ray tomography[41], phase contrast imaging[42], and ptychography[43] are used to obtain missing information about cell morphology. In contrast, the ucKB mirror, which is ideal in terms of broadband energy, focusing efficiency (ratio of focused photons to photons entering the device), and nanofocusing ability, enhances the LEXRF technique by producing a bicolor nanoprobe. It can be used to quantitatively evaluate the specimen morphology and the amount and concentration distributions of both light elements and metal elements

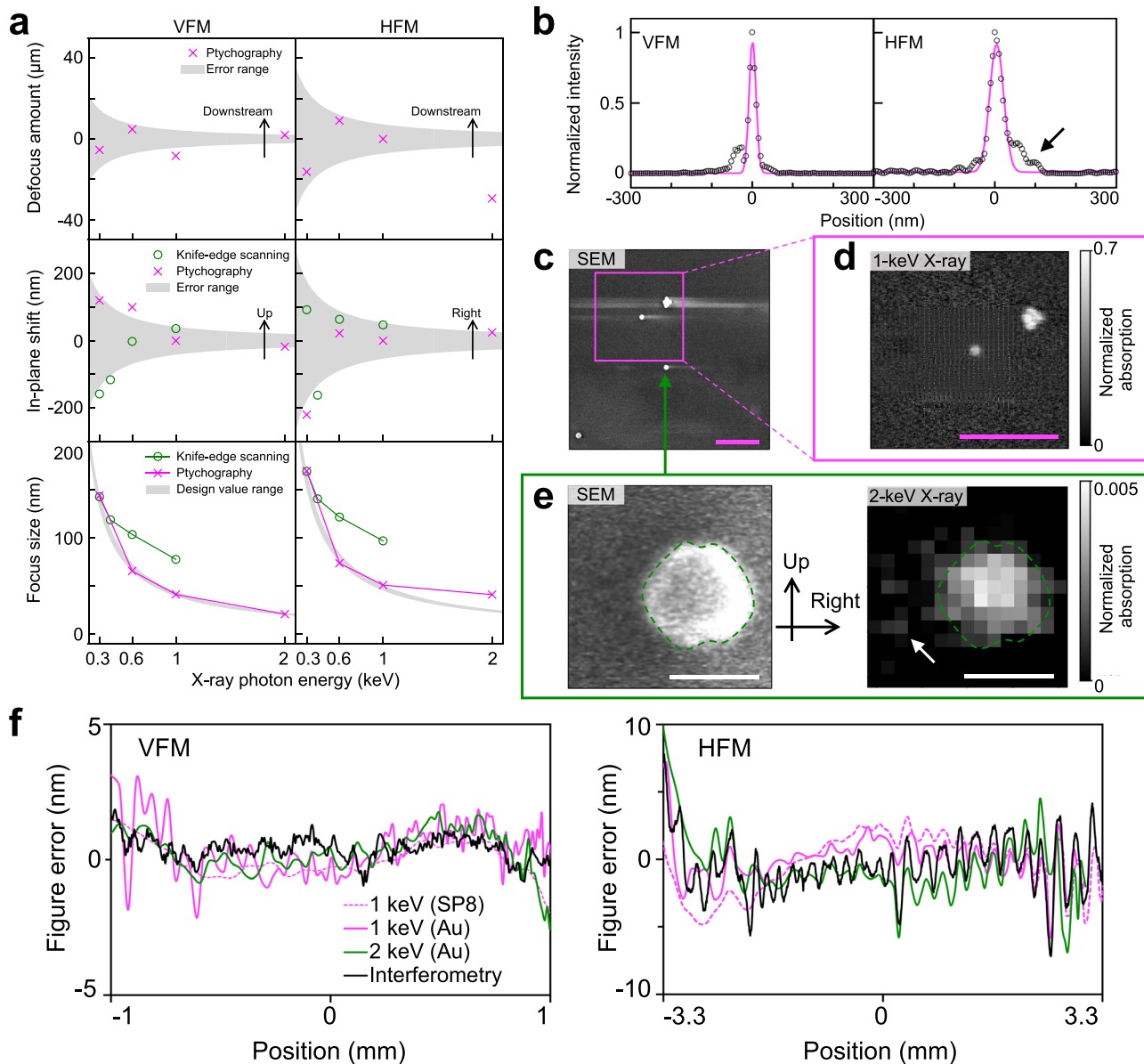

**Fig. 3 | Achromatic soft-X-ray nanoprobe formed by ultracompact Kirkpatrick-Baez mirror. a** Focus position and full width at half maximum (FWHM) focus size versus X-ray photon energy. The error ranges for the defocus amount and the in-plane shift were calculated based on the ± Rayleigh range and ± FWHM focus size, respectively. The knife-edge scanning and ptychography methods both reveal that the increased photon energy condensed soft X-rays into a smaller focus spot. **b** Ptychographically reconstructed focusing profiles with an FWHM spot size of 20.4 nm × 40.7 nm at 2-keV photon energy. **c** Scanning electron microscopy (SEM) and (**d**) ptychography images of Au nanoparticles. The ptychography image was captured at 1-keV photon energy with a 10.3-nm pixel size. The Au particles were mounted on a silicon nitride supporting film. The weak electric conductivity of this film causes charge-up effects, which manifest as a streak in the SEM image. Scale bars in (**c**) and (**d**) 1 $\mu$m. **e** SEM micrograph and 2-keV scanning transmission X-ray microscopy (STXM) image of single Au nanoparticle. The bright area differs in the images as SEM detects electron scattering from the surface and STXM measures the transmittance. The dashed lines highlight the outline of the nanoparticle. The sidelobe in (**b**) causes satellite signals (see arrows). Scale bar: 100 nm. The X-ray images in (**d**, **e**) were captured at the identical focus position. The positive coordinates follow the directions in Fig. 1. **f** Comparison of figure errors converted from ptychographic probe functions and observed using interferometry. The specimens given in parentheses are described in Methods. With the positional origin at the mirror center, the positive x-direction is toward the downstream side of the mirror.

with a lateral resolution of 100 nm. Such comprehensive information has not readily been accessible. The distributions of C, N, and O are similar in Fig. 5e–h and Fig. 6d–f. The C and N concentrations could stem from extracellular matrices or the cytoskeleton, whereas the O concentration could stem from nucleic acids. The unfilled granules in Fig. 5d and h could result from symmetric biological granules found in chemically-fixed samples[44–46], as some labile elements such as K, Ca, and P can leach into the fixation agent[47,48]. Neuronal activity such as transitional spine elimination could explain the Cu and Fe concentration at the bump area shown in Fig. 6c and f[35].

The present study is the first report on multicolor nanoscale X-ray fluorescence microscopy. Various chemical mapping methods are currently being studied to understand multifaceted biological phenomena[49]. LEXRF with an achromatic nanoprobe is expected to be a powerful subcellular mapping tool. For instance, the presented LEXRF can trace F in fluorodeoxy-D-glucose at subcellular resolution for revealing unknown glucose uptake within the brain[50] or cancer cells.

Table-top broadband X-ray sources, such as attosecond lasers, might require even shorter mirrors, which are compatible with the

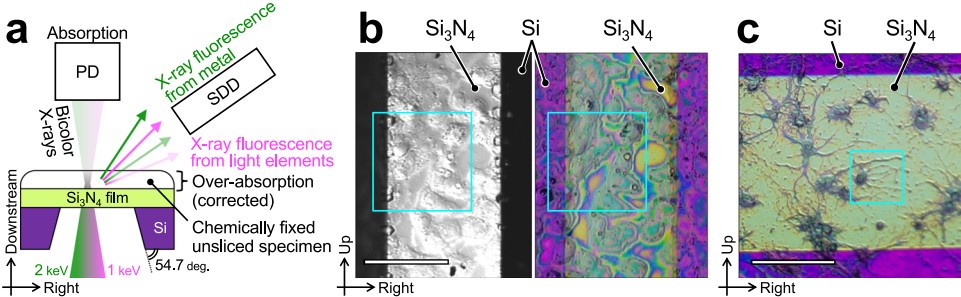

**Fig. 4 | Schematic diagram of low-energy X-ray fluorescence technique based on ultracompact KB (ucKB) mirror. a** Schematic top view of specimens and silicon nitride supporting film window in forward-scattering configuration. The bicolor X-ray nanoprobe is generated by the ucKB mirror. The lower-energy X-rays induce fluorescence from light elements, whereas the higher-energy X-rays induce fluorescence from metals. X-ray fluorescence from the specimens enters the silicon drift detector (SDD). The intensity of the transmitted X-rays is concurrently recorded with the photodiode (PD). Visible-light micrographs of (**b**) Hep G2 and (**c**) primary hippocampal neurons. The monochrome image was captured using differential interference contrast microscopy. The boxed insets show the area observed using X-rays. Scale bar: 100 $\mu m$.

short-focal-length strategy. An extremely short focal length requires a reduced mirror length, making the spatial wavelength of figure errors $d_m$ close to the mirror length $l_m$ in Eq. (1). The employed fabrication method necessitates $r > 0.1$ mm, and thus

$$\Delta u \approx \frac{\lambda r}{d_m \theta_0} \approx \frac{\lambda r}{l_m \theta_0} > \frac{\lambda}{\theta_0} \times \frac{0.1 \text{mm}}{l_m} . \tag{2}$$

$\lambda$ and $\theta_0$ determine the diffraction limit. The shortest mirror length for a small focus size is 0.1 mm. Microscale XRF analyses have been conducted with table-top broadband X-ray sources[51]. The combination of an ucKB mirror and a table-top X-ray source could expand the use of scanning X-ray nanoanalyses, which have been restricted to SR-based X-rays.

## Methods

### Ethical statement

This research complies with all relevant regulations. All animal experiments were performed with the approval of the Animal Experiment Ethics Committee of the University of Tokyo and followed the University of Tokyo Guidelines for the Care and Use of Laboratory Animals.

### Ultrashort focal length and mirror length

Figure 7 compares short and long mirrors with figure errors for a given spatial wavelength. Figure 7a shows a wavefield traveling along the positive $z$-direction from the aperture on the $x$-axis to the focus area across the $u$-axis. An elliptical X-ray mirror crops the incident wavefield into an aperture shape and the surface imperfection disturbs the wavefront. The grazing angle of X-ray mirrors is approximately constant around $\theta_0$. As illustrated in the inset in Fig. 7a, a periodic figure error with amplitude $h_0$ and spatial wavelength $d_m$ is thus assumed to be replicated to the aperture wavefront with height error $h_a = 2h_0 \sin \theta_0$ and spatial wavelength $d_a = d_m \sin \theta_0$[52]. The wavefront at the aperture $U_a$ modulated by the wavefront error $P_m$ can be modeled as

$$
\begin{aligned}
U_a^{error}(x) &= P_m(x) U_a^{ideal}(x) \\
&\approx \exp\left[ j \frac{2\pi}{\lambda} h_a \cos\left(\frac{2\pi}{d_a} x\right) \right] U_i(x) \text{rect}\left(\frac{x}{a}\right) ,
\end{aligned}
\tag{3}
$$

where $2a$ is the aperture width, $\lambda$ is the wavelength of the monochromatic light, $U_i$ is the incident wavefield, which is normally uniform, and rect($x/a$) returns non-zero values under $|x| \le a$.

The wavefield circularly transformed by the focusing device follows the Fraunhofer diffraction formula[53]. Provided that $h_0$ is significantly small and the NA is constant at $\alpha$ (arctan($a/r$) $\approx a/r \approx \alpha$), the focused wavefield can be expressed as

$$
\begin{aligned}
U_f(u) &= A(u;r) r \alpha \left[ \mathcal{F}[P_m U_i]\left(\frac{u}{\lambda r}\right) * \text{sinc}\left(\frac{\alpha u}{\lambda}\right) \right] \\
&\approx A(u;r) r \alpha \left\{ (1 - \phi^2) \text{sinc}\left[\frac{\alpha u}{\lambda}\right] \right. \\
&\quad - \frac{\phi^2}{2}\left( \text{sinc}\left[\frac{\alpha}{\lambda}\left(u - \frac{2\lambda r}{d_a}\right)\right] + \text{sinc}\left[\frac{\alpha}{\lambda}\left(u + \frac{2\lambda r}{d_a}\right)\right] \right) \\
&\quad \left. + j\phi\left( \text{sinc}\left[\frac{\alpha}{\lambda}\left(u - \frac{\lambda r}{d_a}\right)\right] + \text{sinc}\left[\frac{\alpha}{\lambda}\left(u + \frac{\lambda r}{d_a}\right)\right] \right) \right\} \\
A(u;r) &= \frac{1}{j\lambda r} \exp\left[ j\frac{\pi u^2}{\lambda r} \right] ,
\end{aligned}
\tag{4}
$$

where $U_f$ is the focus wavefield, $\mathcal{F}$ is the Fourier transform operator, the symbol * denotes the convolution operation, $\phi = \pi h_a / \lambda$, and $r$ is the distance between the focus and the downstream mirror end.

Five normalized sinc functions underlie $U_f$. The first term is independent of $r$ and centered at $u = 0$ (resulting from the ideal focus); it shows that smooth mirrors with the same NA form an identical focus spot regardless of focal length. The other terms are centered either at $u = \pm \lambda r / d_a$ or $\pm 2\lambda r / d_a$, whose positions diverge with $r$. They express extra focus spots diffracted by surface imperfections. The gap between the extra focus spots $\Delta u$ is

$$\Delta u = \frac{\lambda r}{d_a} = \frac{\lambda r}{d_m \sin \theta_0} \approx \frac{\lambda r}{d_m \theta_0} , \tag{5}$$

as described in Eq. (1). Figure errors of $d_m = 500$ $\mu m$ were observed during mirror production. The soft-X-ray region spans about $\lambda = 1.24$ nm (1-keV photon energy). A reflectivity of 50% restricts $\theta_0$ to approximately 25 mrad or less[54]. Access to the mesoscopic scale requires $\Delta u = 100$ nm. Substituting these parameters into Eq. (5) yields $r \approx 1$ mm.

The advantage of a short focal length was examined using an in-house wave propagation code based on the Fresnel-Kirchhoff diffraction formula[55]. Large figure errors were randomly generated under the assumption that their spatial frequency spans a range of 500 to 2000 $\mu m^{-1}$. Such errors can remain after mirror fabrication. To simulate fabrication-constrained mirrors, the amplitudes of the figure errors were increased beyond the range determined by Rayleigh's quarter wavelength rule, as shown in Fig. 7c. Then, the focusing profiles generated by mirrors with different focal lengths were calculated with the figure errors fed into the simulation. The mirror length was adjusted to achieve the same NA (0.0149) with different focal lengths. Figure 7d shows that short-focal-length

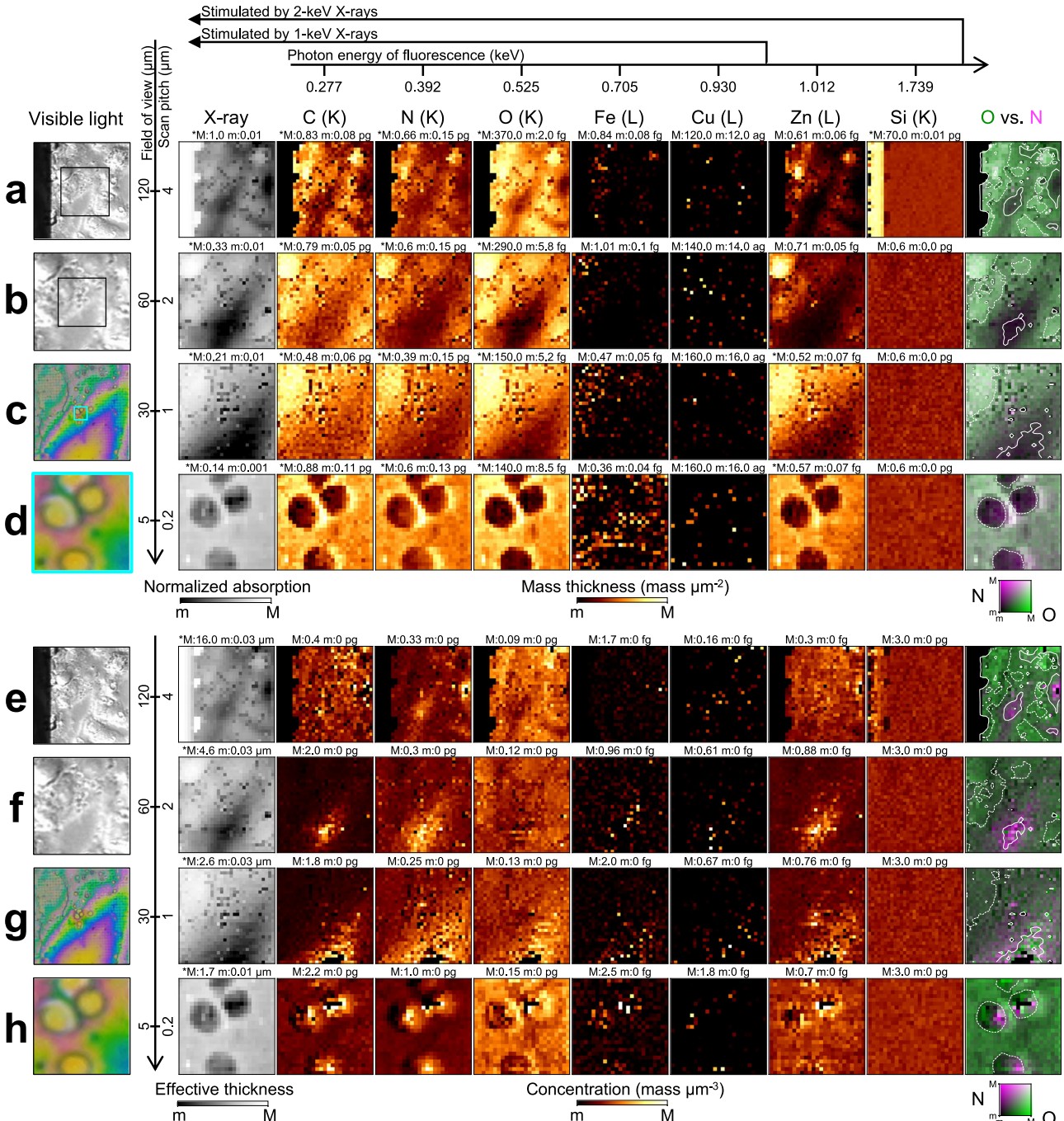

**Fig. 5 | X-ray fluorescence and absorption for Hep G2 cells detected by low-energy X-ray fluorescence technique. a–d** Annotated visible-light micrographs, X-ray absorption images, and X-ray fluorescence images. The boxed inset in the visible-light image shows the field of view adopted in the following close-up image. The fluorescence images are in units of mass thickness (mass $\mu m^{-2}$). The black area in the visible-light image in (**a**) corresponds to the Si window frame, which is almost opaque to X-rays. Cell culturing remnants are attached to the Si frame at the back of the supporting film. They appear as structures projecting from the window frame, significantly reducing the incident X-ray intensity. **e–h** Plain visible-light

micrographs, effective thickness, and X-ray fluorescence images for the same area as that shown in (**a–d**). The fluorescence images are in units of mass concentration (mass $\mu m^{-3}$). The maximum (M) and minimum (m) values for the range are given above each image and the asterisk (*) indicates a logarithmic scale. The electron shell whose vacancy is filled for fluorescence is given in parentheses next to each chemical element. Monochrome visible-light images were captured using the differential interference contrast microscope whereas color ones were observed with the digital microscope.

mirrors can generate a clear main peak, especially with large figure errors.

Even though post-production simulation can be used to numerically evaluate the effect of figure errors on focusing performance[52,55], the short-focal-length strategy can remove fabrication restrictions in advance, as shown in Fig. 7b-d. A detailed mathematical derivation is provided in Supplementary Section 1.

## Mirror fabrication

The figure correction employed an upscaled dynamic stencil deposition technique, which was originally developed to produce simple nano- or microscale deposition patterns through shadow masks[56]. A scanning white-light interferometer (SWLI) was used for the metrology.

Cylindrical substrates were formed with the ROC values listed in Table 1. They were commercially polished (Natsume Optical Corp.).

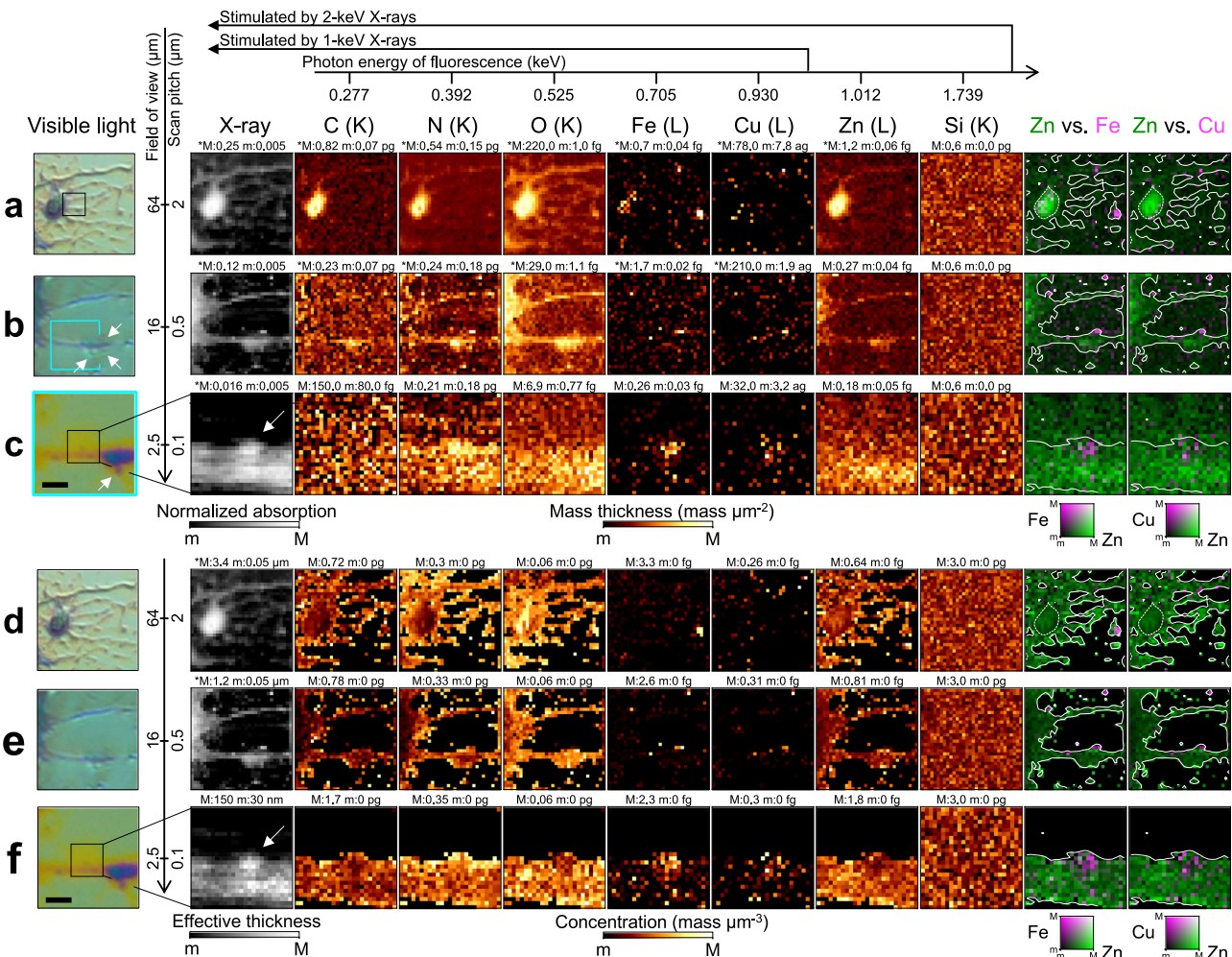

**Fig. 6 | X-ray fluorescence and absorption for primary hippocampal neurons detected by low-energy X-ray fluorescence technique. a–c** Annotated visible-light micrographs, X-ray absorption images, and X-ray fluorescence images. The boxed inset in the visible-light image shows the field of view adopted in the following image. The fluorescence images are in units of mass thickness (mass $\mu m^{-2}$). **d–f** Plain visible-light micrographs, effective thickness, and X-ray fluorescence images for the same area as that shown in (**a–c**). The fluorescence images are in units of mass concentration (mass $\mu m^{-3}$). The arrow in the absorption image indicates a bump area in (**c–f**). The scale bar in (**c–f**) corresponds to 2 $\mu m$. The maximum (M) and minimum (m) values for the range are given above each image and the asterisk (*) indicates a logarithmic scale. The shell whose vacancy is filled with electrons for fluorescence is given in parentheses next to each chemical element.

The super-polished substrates were cut into an L-shape composed of ballast and mirror parts. The ballast part had flat sides and bottoms, enhancing positioning reproducibility during the fabrication process and preliminary mechanical alignment before focusing experiments. The thick deposition on the cylindrical mirror part was performed with coarse and fine figure corrections. Dynamic-stencil-based deposition was used for threefold figure correction. We also developed a double-stencil method[57] for fine figure correction.

We developed single-aperture acquisition and static microstitching interferometry (sMSI) approaches for mirror metrology that use an SWLI (NewView 700S, Zygo)[57]. These interferometry approaches utilize a 2.5X objective lens with an FOV of 3.5 mm × 2.8 mm and has a vertical scanning range of 20 $\mu m$. Conventional microstitching interferometry (MSI) limits the effective FOV to the area where the mirror surface is at the exact focus of the objective lens and few interference fringes appear[13]. This restriction on the FOV greatly increases the required number of stitched frames for highly curved mirrors. To utilize a large FOV, the developed measurement method scans the objective lens vertically. The mirror surface is placed within a long scanning range and the translated focus allows for mirror measurement over the entire FOV. Mirror motions during measurement are limited to only lateral translation, in contrast to the rotation and translation for conventional MSI. Before sMSI generates the mirror surface data, the height offsets between neighboring sub-apertures are adjusted and their overlaps are simply blended. Conventional MSI requires more than a hundred frames[13]. With our efficient metrology approaches, ultracompact mirrors can be measured with a single frame or no more than 10 stitched frames. This evaluation is less error-prone than existing methods.

### Soft-X-ray high-resolution mapping system

The ucKB mirror was mounted on the SHIMA manipulator with 15 piezo-driven stages for the VFM, HFM, sample, and entrance slits. These four modules were installed within the manipulator dimensions of 138 mm × 133 mm × 250 mm[54]. To demonstrate the ucKB mirror for X-ray microscopy, an X-ray charge-coupled device camera (2048 × 2048 pixels, pixel size: 13.5 $\mu m$ × 13.5 $\mu m$, PIXIS-XO:2048B, Princeton Instruments), a photodiode, and a silicon drift detector (XR-100SDD, Amptek) were installed downstream, as illustrated in Fig. 1a[58]. The forward-scattering configuration blocks the noise scattered from the mirror surface. All detectors and the SHIMA manipulator were under vacuum in the SHIMA chamber. The SHIMA system can synchronize all components.

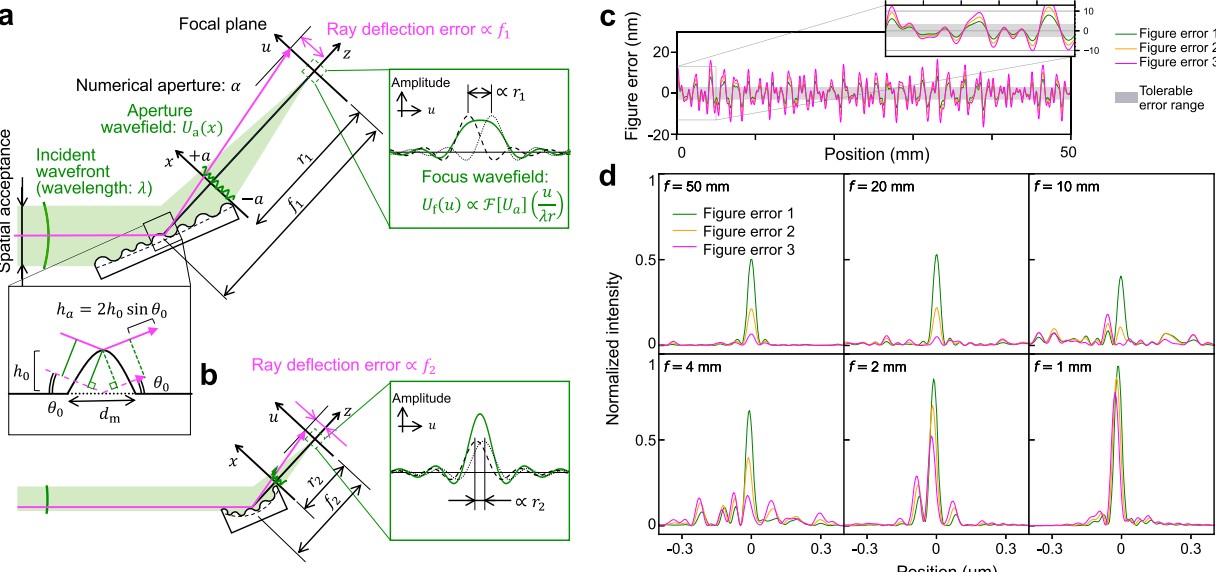

**Fig. 7 | Ray and wavefield behavior for grazing-incidence mirrors with long and short focal lengths coupled with large and small radii of curvature respectively, for achieving given numerical aperture. a** Schematic rays and interference fringes magnified by long focal length of uneven mirror. The inset at the mirror surface illustrates that the figure errors influence the aperture wavefront by a factor of $\sin\theta_0$. **b** Reduced focal spread using short-focal-length mirror whose figure errors exist at same spatial wavelength as that of long-focal-length mirror. **c** Figure errors input to wave-optics simulation. The figure errors were extracted in a position range of zero to the mirror length. Figure error 1 (peak-to-valley and rootmean-square errors: 16.4 and 3.0 nm, respectively) was generated by summing random sinusoidal curves in the frequency range of 500 to 2000 $\mu m^{-1}$. The amplitudes were increased by factors of 1.5 and 2 for Figure errors 2 and 3, respectively. The tolerable error range is ± 3.1 nm, which was calculated based on Rayleigh's quarter wavelength rule for 1-keV X-rays. **d** Focusing profiles at 1-keV photon energy depending on focal length and figure error. The intensity is normalized by the ideal peak intensity.

The incident path length is different between the VFM and the HFM. This configuration stems from the motion reliability required for the secondary slits, which can be achieved using independent vertical and horizontal slits at the beamline. The slits are located downstream of the gratings at beamline BL25SU-A (see Fig. 1a).

### Sample specifications

During the mechanical alignment, two types of in-house test specimen were installed at the designed focus position. The details are given in Supplementary Section 2.

The first type of test specimen included a 10-$\mu$m-diameter pinhole for knife-edge scanning, a pattern for ptychography with the letters "SP8" spanning an area of $2\,\mu m \times 3\,\mu m$, and scattered $\phi$100-nm Au nanoparticles. All specimens were produced on a transmission electron microscopy (TEM) grid (PELCO Silicon Nitride Support Films for TEM, TED PELLA, Inc.), which was a 3-mm-diameter Si disk with nine 200-nm-thick silicon-nitride supporting film windows. The isolated single Au nanoparticles shown in Fig. 3e had vertical and horizontal dimensions of 113.1 and 129.0 nm, respectively.

The second type of test specimen included air-dried biological cultures[37,59] for LEXRF microscopy. The specimens were cultured on in-house Si wafer chips with 200-nm-thick silicon-nitride supporting film windows.

### Focusing experiment

Before the SR experiments, the ucKB mirror was mechanically adjusted in the SHIMA system by observing the mirror distance and posture with visible-light microscopes. The ucKB mirror was thus roughly aligned with the primary axis of the SHIMA system, which is approximately identical to the incident X-ray path. During the SR experiments, grating-mirror pairs M21a-G3a and M21a-G4a were selected for the soft-X-ray ranges of 0.3 to 1 keV and 0.4 to 2 keV, respectively. The center groove density was 300 lines/mm for M21a-G3a and 600 lines/mm for M21a-G4a. The latter grating was used for STXM and LEXRF

microscopy. The geometrical focus size was controlled to one-tenth of the diffraction limit. The secondary vertical slit opening (S2) was set to 180, 135, 54, and 27 $\mu$m for photon energies of 0.3, 0.4, 1, and 2 keV, resulting in energy resolutions of $E/\Delta E = 1300$, 2200, 1700, and 15000, respectively. The corresponding horizontal openings were 45, 33, 13.5, and 6 $\mu$m at the secondary horizontal slit (SCa), respectively. The ucKB mirror was finely adjusted to the incident X-rays using a Foucault knife-edge test with the 10-$\mu$m-diameter pinhole. In the Foucault knife-edge test, the lowest photon energy available for the grating-mirror pair was initially used because this energy is less penetrative against the Ni-coated specimens and thus the transmitted X-rays exhibit clearer responses to the knife edge. Then, the energy was increased to 1 keV, which reduced the depth of focus by nearly one-third. The higher energy allowed the specimen to be finely positioned along the X-ray path. The SHIMA system basically maintained the same mirror posture unless the mirror-grating pair was changed. The readjustment of the ucKB mirror achieved an approximately identical focus size at 1-keV photon energy. Relaying the focus spot at this energy does not compromise the integrity of the achromatic nanofocusing. Because of the upstream optics involved, the focus position can be slightly shifted, as shown in Fig. 3a.

### Ptychography

Ptychography is a computational imaging technique that reconstructs the probe illumination function and sample transmittance function through an iterative calculation process. Multiple intensity images that capture overlapping sample areas provide additional information for obtaining optimal results[60]. The large-NA design of the ucKB mirror leads to a narrow depth of focus, which becomes very narrow at 2-keV photon energy. Because a knife edge as thin as this depth of focus cannot effectively attenuate 2-keV soft X-rays, the focusing profiles were ptychographically retrieved[61-63] using the extended ptychographical iterative engine (ePIE) algorithm[64]. The in-house test specimens were translated at the focus spot in a raster scanning path. The

energy resolution was unchanged from that in the focusing experiments to maintain lateral coherence. The step size was smaller than the focus size to produce beam overlaps. The exposure time per image was typically 1 s except for 2-keV photon energy; a glass plane mirror upstream of the ucKB mirror caused a low photon flux at 2 keV, leading to an exposure time of 7 s.

Ptychography can reconstruct the wavefront of a focused X-ray beam, which reflects figure errors on a mirror surface[19,57,61]. The probe function was first reconstructed and propagated to the downstream side of the mirrors. The propagated wavefront was then converted to figure errors. The results indicate that the mirror surface illuminated by X-rays was slightly shifted lengthwise depending on the photon energy. Because such shifts modify the optimal ellipse-fitting parameters, Fig. 3a shows the range of theoretical diffraction-limited focus sizes derived from the varying ellipse parameters.

Ptychography has become a de facto standard for precisely evaluating X-ray nanobeams because its iterative calculations empirically give unique solutions for sample transmittance and probe function[61–63]. However, a previous study reported that although the converged calculation successfully recovers the sample transmittance, the probe function underestimates the focus size of a strongly diffused nanobeam[18]. This could stem from high-frequency figure errors and surface roughness, which produce complex reflected X-ray images with structures smaller than the detector pixels. The ptychography results reported in the present study are well grounded in the figure error evaluated using the interferometer. See Supplementary Section 2 for detailed figure error reconstruction.

### Low-energy X-ray fluorescence microscopy and analysis

The energy resolution was detuned to approximately $E/\Delta E = 800$ to increase the incident photon flux. The vertical and horizontal slit openings were 200 and 430 $\mu$m, respectively, theoretically producing a geometrical probe size of 20 nm × 160 nm at the focus. The probe was defocused for a large scanning pitch by moving the specimen away from the focus. The horizontal slit opening was narrowed for a smaller probe size. The peaking time of the SDD was 25.6 $\mu$s. The specimens were moved in a raster scanning path with an exposure time of 3 s.

The X-ray fluorescence spectra recorded at each scanning point contained both X-ray fluorescence peaks generated by chemical elements and elastic scattering peaks. To separate them, the spectra were Gaussian-fitted using the least-squares method.

Our iterative method can calculate the sample density and thickness. It is based on the ZAF correction method (Z: atomic number, A: absorption, F: fluorescence), which is well established for quantitative analyses using energy-dispersive X-ray spectroscopy with electron probes[65]. The photon flux for the incident X-rays was $10^5$ times greater than the amount of primary X-ray fluorescence, which generates secondary X-ray fluorescence. Therefore, among the ZAF corrections, the present study corrected only the absorption effect of the incident X-rays and X-ray fluorescence (over-absorption) based on the Lambert–Beer law. Mass ratios of chemical elements, average mass absorption coefficient and sample density, and sample thickness were repeatedly calculated at each scanning point. A uniform chemical element concentration was assumed along the thickness direction at each scanning point. The fluorescence counts were converted into mass thickness using the fundamental parameter method. The recorded signals for X-ray absorption and fluorescence were assumed to be the sum of signals individually triggered by the two photon energies. The intensity ratio of the incident 2-keV X-rays to 1-keV X-rays was calculated using fluorescence from the known quantities of N and Si in the supporting film. The sum of the intensity was measured using the photodiode. The intensity of the X-ray fluorescence from the known quantities of N and Si in the supporting film was also used to calibrate the fluorescence from other elements induced by 1- and 2-keV X-rays. The solid densities of C, N, and O within the specimens were based on their respective average densities in the essential amino acids.

### Reporting summary

Further information on research design is available in the Nature Portfolio Reporting Summary linked to this article.

## Data availability

The data that support the findings of this study can be found in the article and the Supplementary Information file. Any other relevant data are available from the corresponding author upon request. Source data are provided with this paper.

## Code availability

The analysis and simulation codes are available from the corresponding author upon request.

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

## Acknowledgements

We would like to sincerely thank Dr. Yoshinori Kotani and Dr. Kiyofumi Nitta from the Japan Synchrotron Radiation Research Institute (JASRI) for supporting the SDD operations at SPring-8. We would also like to express our gratitude to Dr. Makina Yabashi (RIKEN) and Mr. Kai Sakurai (The University of Tokyo) for a fruitful discussion and for providing Si wafer chips with silicon nitride membrane windows, respectively. We acknowledge the Japan Society for the Promotion of Science (JSPS) KAKENHI for its support under grant numbers JP20J21562 (T.S.), JP21K20394 (Y.T.), JP20H04451 (T.K.), JP20K20444 and JP23H00156 (H.M.). T.S. is financially supported by the DC1 Research Fellow Program of the JSPS. The SR experiment was performed at beamline BL25SU-A at SPring-8, Japan, with the approval of JASRI (proposal nos. 2021B1836 and 2021A1612). Part of this work was conducted in the Research Hub for Advanced Nano Characterization, The University of Tokyo, supported by the Ministry of Education, Culture, Sports, Science and Technology (MEXT), Japan, under grant numbers JPMXP09A19UT0306 and JPMXP09A20UT0389. Part of this work was also supported by "Nano-technology Platform Japan" of MEXT under grant JPMXP09F21UT0117. Part of this work was conducted at Takeda Clean Room / Ultrafine Lithography and Analysis Center of The University of Tokyo, supported by "Advanced Research Infrastructure for Materials and Nanotechnology in Japan" (ARIM) under grant JPMXP1222UT1008.

## Author contributions

T.S. and M.H. conceived the mirror design. T.S. wrote the in-house C-based wave propagation code. T.S. fabricated the mirrors and the in-house test specimens. T.S., Y.T., T.K., Y.S., H.K., H.O. and H.M. planned the focusing experiments, built the SHIMA system, and conducted the SR-based focusing experiments. Y.T. wrote the C++ ptychography source code. T.S. and Y.T. analyzed the results of the focusing experiments. T.S. conceived the LEXRF technique using bicolor soft X-rays and the reconstruction of sample thickness. T.S., Y.T., F.M., T.K., M.S., Y.S., H.K., H.O., K.S., Y.J. and H.M. planned the LEXRF experiments. F.M. cultured the primary hippocampal neurons. M.S. cultured the Hep G2 cells. T.S., Y.T., F.M. Y.S, H.K. and H.O. performed the SR experiments for LEXRF. T.S. wrote the Python source code for fluorescence analyses and performed multi-Gaussian fitting, ZAF correction, and visualization. T.S., F.M., M.S., K.S., and Y.J. interpreted X-ray fluorescence data. T.S., F.M., and M.S. wrote the first draft of the manuscript. All authors read and agreed to the final version of the manuscript.

## Competing interests

The authors declare no competing interests.
