## [Peer Review File · Nature Communications]

Ultracompact mirror device for forming 20-nm achromatic soft-X-ray focus toward multimodal and multicolor nanoanalysesReviewers' Comments:

Reviewer #1:

Remarks to the Author:

The manuscript describes a set of ultrashort KB mirrors (the shortest one is about 2 mm in length and 0.160 m central radius of curvature) from the design, the fabrication, the metrology, and the implementation for soft X-ray nano-focusing (20 nm at 2 keV) and the application for experiments with the focused beam. The work is impressive in many aspects and the manuscript is well organized in general. I would like to recommend the publication of this work if the authors can respond to the following comments or concerns. Here are my detailed comments after reading this manuscript.

1. Lines 089-093. The authors mentioned that soft X-ray nano-focusing requires a larger NA than hard X-ray nano-focusing. In the soft X-ray region, the grazing angle is relatively larger. It makes the focusing wavefront more susceptible to mirror figure errors. The statement itself sounds fine. However, if we discuss focusing with the diffraction limit, we need to consider the wavelength as well. Compared to hard X-ray, the soft X-ray wavefront is more susceptible to the figure error due to the larger grazing angle ($\sim 10\times$), but in the meantime, the tolerance of the wavefront error is also larger due to the longer wavelength ($\sim 10\times$). By telling this full story, the readers can easily understand the experiment result mentioned in lines 098-101: "The focus size does not shrink, but instead expands, with increasing photon energy."
2. Lines 110-112. This sentence is a bit confusing. Could the authors re-phrase this sentence and explain why the evaluation is only in the middle- to short-frequency range (1 cm^{-1} to $10\ \mu\text{m}^{-1}$)? By the way, I suggest the authors not use the term "short-frequency", but "high-frequency" instead.
3. Lines 115-117. The author claims that "Millimeter-scale focal lengths would enhance focusing robustness as X-rays scattered by mirror slope errors reach the focal plane before widely spreading." I believe this is viewed from the partially coherent condition. If with a fully coherent beam, the diffraction-limited focusing will depend on the height errors instead of the slope errors.
4. Section 2.1 and Figure 1. It will be much better to have some wavefront propagation simulation to illustrate what has been presented in Fig. 1. I suggest the authors add a simulation using wavefront propagation.
5. Lines 375-379. These descriptions need to be improved, as I feel it is difficult to understand the exact meaning the authors want to deliver, such as "an unprecedented reflective focus size", "level off" ...
6. Lines 404-405. "... which is small enough to prevent X-ray diffusion." Do the authors want to mean "X-ray scattering"?
7. Lines 405-406. Please explain why authors can conclude that "the uckB mirror can concentrate X-rays into sub-50-nm spots without degrading the incident coherence." Based on the observation that "the interferometry and ptychography results agree within 3 nm." What do the authors mean by mentioning "no degrading coherence"?
8. Section 2.4, Fig. 6, and Fig 7. Elements, like C, N, O, and Fe, can be stimulated by both 1keV and 2keV X-rays. Is it possible in practice to separate them as quantitative results without getting the crosstalk influence between the two energy X-rays?
9. Section 4.1. For mirror metrology, authors developed the single-aperture acquisition method and the sMSI method. In the published Ref. [56], we can see their results have discrepancies in the fitted ellipse parameters (especially the grazing angles). How much tolerance of grazing angle error could this KB mirror afford? Which result (single-aperture or sMSI) do the authors trust more in your

fabrication, and what is the reason for that?

10. Lines 739-743. This paragraph is a surprise. It should not be placed in the mirror fabrication section.

11. Section 4.2. It is better to have a sketch of the beamline here.

12. Line 820. How the ptychography reconstructed probe function be converted to the figure error?

Reviewer #2:

Remarks to the Author:

The manuscript presents a the design, manufacture, and use of the ultracompact Kirkpatrick-Baez (ucKB) mirror that can focus soft x-ray light to spots of 20 nm by 20 nm over a large photon energy range. This result would be very useful in the fields of optics and for future experiments at X-ray sources, like FELs and synchrotrons, and the results shown here from the experiment are convincing in the new mirror's use.

However, the paper needs a large amount of work. Though the subject of the paper is both interesting and relevant, the approach and writing of the paper is hard to follow, meandering, and often does not give the reader all the information one needs. I listed the issues that I found below:

1) The overall quality of the sentence structure in the mirror manufacture and theory section of the paper could be better. Lots of sentences are set conditionally without reason. A good example is in line 105, where the authors say that novel strategies 'might' have to be developed. Considering that the paper is presenting a novel development, this 'might' is pointless. One should say 'have to be developed' or 'have been developed'.

2) The introduction of the paper gives the impression that the paper is theoretical, about how to build the ucKB. It is only from section 2.2 onward that the reader realizes that the mirror is manufactured and an experiment with it is done. The introduction should more clearly state that the authors have design the ucKB, manufactured it, and did an experiment with it that showed new results.

3) The results section has equations and modeling. This is very odd for a results section, since I would expect the theoretical consideration of the mirror to be in the 'setup' or 'experiment section'. To that point, the paper has neither of these sections, which is not something one sees very often. I would strongly recommend the authors re-consider the order and the headings of the sections of the manuscript.

4) The performance in section 2.3 and the figure 4 do not answer some important questions. For example, the authors, in the introduction, mention that one of the major issues with current solutions for soft X-ray focusing leads to energy scans not being easily doable due to the changing position of the focus with photon energy. However, there is no discussion whether the focus stays stable with the ucKB, and no data showing the focus staying on point over different energies in figure 4. They should have a graph similar to 4a for the X, Y, and Z position of the focus as a function of photon energy. The supplementary information provides good information, but an overview graph should be put in the main article to show how the mirror is superior to the previous solutions in every way mentioned in the paper.

If these points are addressed, I think the paper is worth publishing in Nature Communications.

Responses to comments from Reviewer 1

The manuscript describes a set of ultrashort KB mirrors (the shortest one is about 2 mm in length and 0.160 m central radius of curvature) from the design, the fabrication, the metrology, and the implementation for soft X-ray nano-focusing (20 nm at 2 keV) and the application for experiments with the focused beam. The work is impressive in many aspects and the manuscript is well organized in general. I would like to recommend the publication of this work if the authors can respond to the following comments or concerns. Here are my detailed comments after reading this manuscript.

We are grateful for the recognition of the potential of our mirror and its application.

Introduction

C1.1	Lines 089-093. The authors mentioned that soft X-ray nano-focusing requires a larger NA than hard X-ray nano-focusing. In the soft X-ray region, the grazing angle is relatively larger. It makes the focusing wavefront more susceptible to mirror figure errors. The statement itself sounds fine. However, if we discuss focusing with the diffraction limit, we need to consider the wavelength as well. Compared to hard X-ray, the soft X-ray wavefront is more susceptible to the figure error due to the larger grazing angle (~10x), but in the meantime, the tolerance of the wavefront error is also larger due to the longer wavelength (~10x). By telling this full story, the readers can easily understand the experiment result mentioned in lines 098-101: “The focus size does not shrink, but instead expands, with increasing photon energy.”
------	---

As suggested, we have modified the text to explain the tolerance of the wavefront error. Additions are shown in blue.

A1.1	Lines 096-102. Nevertheless, the rigorous fabrication requirements have prevented advanced X-ray mirrors from achieving the ideal soft-X-ray focus size [15–20]. Achromatic soft-X-ray nanofocusing is barely diffraction-limited with a focus size of 241 nm × 81 nm at 0.3-keV photon energy [18]. The focus size does not shrink, but instead expands, with increasing photon energy [19, 20]. Currently available achromatic soft-X-ray nanoprobes are severely limited by the fabrication process used. Lines 096-104. Nevertheless, the rigorous fabrication requirements have prevented advanced X-ray mirrors from achieving the ideal soft-X-ray focus size [15–20]. Achromatic soft-X-ray nanofocusing is barely diffraction-limited with a focus size of 241 nm × 81 nm at 0.3-keV photon energy [18]. The focus size does not shrink, but instead expands, with shortening X-ray wavelength [19, 20] because both the diffraction limit and the wavefront error tolerance for the diffraction limit are proportional to the X-ray wavelength [1]. Currently available achromatic soft-X-ray nanoprobes are severely limited by the fabrication process used.
------	--

C1.2	Lines 110-112. This sentence is a bit confusing. Could the authors re-phrase this sentence and explain why the evaluation is only in the middle- to short-frequency range (1 cm ⁻¹ to 10 μm ⁻¹)? By the way, I suggest the authors not use the term “short-frequency”, but “high-frequency” instead.
------	---

As suggested, we have clarified this sentence and replaced “short-frequency” with “high-frequency”.

A1.2	Lines 110-112. Such mirrors could be precisely fabricated because smooth surfaces and overall freeform shapes must be produced and evaluated only in the middle- to short-frequency range (1 cm⁻¹ to 10 μm⁻¹). Lines 112-115. Such mirrors can be precisely fabricated because only the middle- to high-frequency range (1 cm⁻¹ to 10 μm⁻¹) is crucial to their smooth and freeform shapes (low-frequency figure errors on millimeter-scale mirrors are considered to be linear offsets).
------	--

C1.3	Lines 115-117. The author claims that “Millimeter-scale focal lengths would enhance focusing robustness as X-rays scattered by mirror slope errors reach the focal plane before widely spreading.” I believe this is viewed from the partially coherent condition. If with a fully coherent beam, the diffraction-limited focusing will depend on the height errors instead of the slope errors.
------	--

For clarity, we have replaced “slope error” with “figure error”. The updated phrase (“X-rays reach the focal plane before widely spreading due to mirror figure errors”) describes focusing robustness from the viewpoints of both geometrical optics and wave optics.

A1.3	Lines 114-117. In contrast, short mirrors can bring their foci much closer to the mirror center. Millimeter-scale focal lengths would enhance focusing robustness as X-rays scattered by mirror slope errors reach the focal plane before widely spreading (see Fig. 1). Lines 117-119. In contrast, short mirrors can bring their foci much closer to the mirror center. Millimeter-scale focal lengths enhance focusing robustness as X-rays reach the focal plane before widely spreading due to mirror figure errors (see Fig. 7).
------	--

Results

C1.4	Section 2.1 and Figure 1. It will be much better to have some wavefront propagation simulation to illustrate what has been presented in Fig. 1. I suggest the authors add a simulation using wavefront propagation.
------	---

As suggested, we have added results from a wavefront propagation simulation. The in-house simulation accepted the figure error and focal length as inputs. The simulation results show that a short-focal-length mirror can generate a stronger main focus peak than that produced with a long-focal-length mirror. This tendency becomes obvious for a large figure error. The sidelobes around the center of the focus combine into the main peak as the focal length becomes shorter. Figure 1 in the original manuscript is now Fig. 7 in the Methods section in the revised manuscript (following a suggestion by Reviewer 2).

Fig. 1 Schematic comparison of ray and wavefield behavior for grazing-incidence mirrors with long and short focal lengths coupled with large and small ROCs respectively, for achieving given NA. a Rays and interference fringes magnified by long focal length of uneven mirror. The inset at the mirror surface illustrates that the figure errors influence the aperture wavefront by a factor of $\sin \theta_0$. b Reduced focal spread using short-focal-length mirror whose figure errors exist at same spatial wavelength as long-focal-length mirror.

A1.4

Fig. 7 Ray and wavefield behavior for grazing-incidence mirrors with long and short focal lengths coupled with large and small radii of curvature respectively, for achieving given numerical aperture. **a** Schematic rays and interference fringes magnified by long focal length of uneven mirror. The inset at the mirror surface illustrates that the figure errors influence the aperture wavefront by a factor of $\sin \theta_0$. **b** Reduced focal spread using short-focal-length mirror whose figure errors exist at same spatial wavelength as that of long-focal-length mirror. **c** Figure errors input to wave-optics simulation. The figure errors were extracted in a position range of zero to the mirror length. Figure error (peak-to-valley and root-mean-square errors: 16.4 and 3.0 nm, respectively) was generated by summing random sinusoidal curves in the frequency range of 500 to 2000 μm^{-1} . The amplitudes were increased by factors of 1.5 and 2 for Figure errors 2 and 3, respectively. The tolerable error range is ± 3.1 nm, which was calculated based on Rayleigh's

quarter wavelength rule for 1-keV X-rays. d Focusing profiles at 1-keV photon energy depending on focal length and figure error. The intensity is normalized by the ideal peak intensity.

C1.5	Lines 375-379. These descriptions need to be improved, as I feel it is difficult to understand the exact meaning the authors want to deliver, such as “an unprecedented reflective focus size”, “level off” ...
------	---

“An unprecedented reflective focus size” was meant to indicate that such a small focus size was for the first time achieved using reflective focusing devices. As *Nature Communications* discourages authors from using this type of phrase throughout the manuscript, we have removed the expression.

Lines 376-378.

Nevertheless, knife-edge scanning at 1-keV photon energy has ~~an unprecedented reflective~~ focus size of 77.4 nm × 96.7 nm in terms of the full width at half maximum (FWHM).

C1.5a Lines 331-334
Nevertheless, knife-edge scanning at 1-keV photon energy has a focus size of 77.4 nm × 96.7 nm in terms of the full width at half maximum (FWHM).

“To level off” in this context was meant to indicate that the measured horizontal focus size stayed higher than the design value at 2-keV photon energy. We have rephrased this expression to be more direct.

Lines 378-381.

The large figure error for the HFM causes the horizontal focus size to ~~level off~~ at 2-keV photon energy.

C1.5b Lines 334-336.
The large figure error for the HFM causes the horizontal focus size to not reach the design value at 2-keV photon energy.

C1.6	Lines 404-405. “... which is small enough to prevent X-ray diffusion.” Do the authors want to mean “X-ray scattering”?
------	--

We have replaced “X-ray diffusion” with “X-ray scattering”. We have also added follow-up information to the description of ptychography in the Methods section.

Our preliminary results show that a reflective surface with a highly granular structure causes incident X-rays to spread around the focus. The highly granular structure stems from the crystallization of the coating material, which has a high surface roughness. Strongly diffused X-ray beams can have a finer structure at the focus than that assumed in ptychography calculations, possibly resulting in a less precise evaluation of the focused beams (see Ref. 18: Takeo 2020a and the description of ptychography in the Methods section). In the revised manuscript, the surface roughness of the mirror was small enough to prevent such X-ray diffuse reflection. The ptychography measurement of such cleanly focused beams is expected to be precise.

Lines 404-405.

The root-mean-square surface roughness in a 500-nm square area of the mirror was 0.43 nm, which is small enough to prevent X-ray ~~diffusion~~.

C1.6a Lines 359-361.
The root-mean-square surface roughness in a 500-nm square area of the mirror was 0.43 nm, which is small enough to prevent X-ray scattering.

Lines 825-833.

Ptychography has become a de facto standard for precisely evaluating X-ray nanobeams because its iterative calculations empirically give unique solutions for sample transmittance and probe function [60–62]. However, a previous study reported that although the converged calculation successfully recovers the sample transmittance, the probe function underestimates the focus size of a strongly diffused nanobeam [18]. The ptychography results reported in the present study are well grounded in the figure error evaluated using the interferometer

Lines 914-925.

C1.6b Ptychography has become a de facto standard for precisely evaluating X-ray nanobeams because its iterative calculations empirically give unique solutions for sample transmittance and probe function [60–62]. However, a previous study reported that although the converged calculation successfully recovers the sample transmittance, the probe function underestimates the focus size of a strongly diffused nanobeam [18]. This could stem from high-frequency figure errors and surface roughness, which produce complex reflected X-ray images with structures smaller than detector pixels. The ptychography results reported in the present study are well grounded in the figure error evaluated using the interferometer. See Supplementary Section 2 for detailed figure error reconstruction.

C1.7	Lines 405-406. Please explain why authors can conclude that “the ucKB mirror can concentrate X-rays into sub-50-nm spots without degrading the incident coherence.” Based on the observation that “the interferometry and ptychography results agree within 3 nm.” What do the authors mean by mentioning “no degrading coherence”?
------	---

These sentences (“the ucKB mirror can concentrate ...” and “the interferometry and ptychography results ...”) show that the ptychography successfully reconstructed the probe functions. The wavefront information at the downstream end of the mirror, which was calculated from the probe function, shows features comparable to the results of the interferometry evaluation over different photon energies. When the ptychography reconstruction is not successful, the focus spot size can be ideal but the wavefront no longer retains such figure error information. The agreement between interferometry and ptychography results supports the reliability of the focusing results. Therefore, we have deleted “without degrading the incident coherence” and state that the focus size was precisely evaluated using ptychography.

Lines 401-409.

The figure errors for VFM and HFM were determined based on the ptychography results, as shown in Fig. 4f. The interferometry and ptychography results agree within 3 nm. The root-mean-square surface roughness in a 500-nm square area of the mirror was 0.43 nm, which is small enough to prevent X-ray ~~diffusion~~. It is thus concluded that the ucKB mirror can concentrate X-rays into sub-50-nm spots ~~without degrading the incident coherence~~. See Methods and Supplementary Section 2 for details of the preparation of test specimens and reconstructed figure errors, respectively.

Lines 355-364.

C1.7a The figure errors for the VFM and HFM were determined based on the ptychography results, as shown in Fig. 3f. The interferometry and ptychography results agree within 3 nm, showing that the wavefields were successfully reconstructed. The root-mean-square surface roughness in a 500-nm square area of the mirror was 0.43 nm, which is small enough to prevent X-ray scattering. It is thus concluded that the focus size was precisely evaluated using ptychography and that the ucKB mirror can concentrate X-rays into sub-50-nm spots. See Methods and Supplementary Section 2 for details of the preparation of test specimens and reconstructed figure errors, respectively.

Regarding “without degrading the incident coherence”, we had two concerns about the evaluation of the focus spot. The first concern was that the surface roughness may deteriorate the coherence of the focused beams. The surface roughness of highly granular surfaces in a 500 nm × 500 nm region can locally reach tens of nanometers. The diffuse reflection scatters the incident X-rays around the focus spot. The coherence around the focus is the average of the coherence of the focused X-rays scattered from the entire mirror surface, resulting in the degradation of coherence if a highly granular surface is used. The second concern was that the highly granular surface may produce very small structures in the focused beam intensity distribution. Such small structures may not be well resolved in the ptychography calculation, as the pixel size determined by the detector-sample geometry can be larger than the size of such small structures. This would result in degraded coherence for the ptychography calculation (intensity distribution exists, but there are no clear diffraction patterns). The results of the surface roughness evaluation for our mirrors indicate that these concerns were unfounded.

We address these concerns in the Methods section.

Lines 825-833.

Ptychography has become a de facto standard for precisely evaluating X-ray nanobeams because its iterative calculations empirically give unique solutions for sample transmittance and probe function [60–62]. However, a previous study reported that although the converged calculation successfully recovers the sample transmittance, the probe function underestimates the focus size of a strongly diffused nanobeam [18]. The ptychography results reported in the present study are well grounded in the figure error evaluated using the interferometer

Lines 914-925.

C1.7b Ptychography has become a de facto standard for precisely evaluating X-ray nanobeams because its iterative calculations empirically give unique solutions for sample transmittance and probe function [60–62]. However, a previous study reported that although the converged calculation successfully recovers the sample transmittance, the probe function underestimates the focus size of a strongly diffused nanobeam [18]. This could stem from high-frequency figure errors and surface roughness, which produce complex reflected X-ray images with structures smaller than detector pixels. The ptychography results reported in the present study are well grounded in the figure error evaluated using the interferometer. See Supplementary Section 2 for detailed figure error reconstruction.

C1.8	Section 2.4, Fig. 6, and Fig 7. Elements, like C, N, O, and Fe, can be stimulated by both 1keV and 2keV X-rays. Is it possible in practice to separate them as quantitative results without getting the crosstalk influence between the two energy X-rays?
------	--

It is indeed possible to separate multiple photon energies in our X-ray fluorescence experiment. The known quantities of Si and N in the silicon nitride membrane were used to calibrate the intensity ratio of the incident X-rays. The K absorption edges of Si and N are located above and below 1 keV, respectively. Therefore, the X-ray fluorescence from Si is stimulated by 2-keV X-rays and that from N is stimulated by 1- and 2-keV X-rays. If each photon energy independently contributes to the generation of X-ray fluorescence, a set of two equations (each for N and Si) can be written for the relationship between the intensity of the incident X-rays and the intensity of X-ray fluorescence. A comparison of the amounts of X-ray fluorescence with the theoretical values can be used to calculate the intensity of the incident X-rays. In our experiment, the intensity of 2-keV photon energy was around 1% or less than that of 1-keV photon energy. Thus, the above approach seems valid. The total intensity of the X-rays was measured using a photodiode.

In the revised manuscript, we have added a description of the separation of the photon energies.

Lines 859-867.

The fluorescence counts were converted into mass thickness using the fundamental parameter method. ~~N and Si in the Si₃N₄ supporting film were used to calibrate the fluorescence induced by 1- and 2-keV X-rays. The ratio of 2-keV X-rays to 1-keV X-rays was also calculated.~~ The recorded signals for X-ray absorption and fluorescence were assumed to be the sum of signals individually triggered by the two photon energies. The

solid densities of C, N, and O within the specimens were based on their respective average densities in the essential amino acids.

Lines 948-959.

C1.8 The fluorescence counts were converted into mass thickness using the fundamental parameter method. The recorded signals for X-ray absorption and fluorescence were assumed to be the sum of signals individually triggered by the two photon energies. The intensity ratio of the incident 2-keV X-rays to 1-keV X-rays was calculated using fluorescence from the known quantities of N and Si in the supporting film. The sum of the intensity was measured using the photodiode. The intensity of the X-ray fluorescence from the known quantities of N and Si in the supporting film was also used to calibrate the fluorescence from other elements induced by 1- and 2-keV X-rays. The solid densities of C, N, and O within the specimens were based on their respective average densities in the essential amino acids.

Methods

C1.9	Section 4.1. For mirror metrology, authors developed the single-aperture acquisition method and the sMSI method. In the published Ref. [56], we can see their results have discrepancies in the fitted ellipse parameters (especially the grazing angles). How much tolerance of grazing angle error could this KB mirror afford? Which result (single-aperture or sMSI) do the authors trust more in your fabrication, and what is the reason for that?
------	--

The ultracompact mirrors have a relatively narrow spatial acceptance by design and they were more precisely fabricated compared to those in Ref. [56]. Therefore, the fitted parameters did not significantly change in the radiated mirror area.

Regarding the grazing angle error tolerance, as long as the mirrors are positioned in the KB geometry, the deviation of the fitted grazing angle from the designed value does not affect the focusing of X-rays. In the parameter fitting process, the grazing angle and the focal length are varied and the incident path length is fixed. As a result, the total path length of the vertically focusing mirror (VFM) and that of the horizontally focusing mirror (HFM) can slightly differ. When such a difference caused an astigmatism around the focus, we slightly shifted the HFM along the incident X-ray path to fill the gap between the vertical and horizontal focus positions. We confirmed that the horizontal focusing profile at the horizontal focus position was almost the same as that at the vertical focus position after this sliding process. Considering the manipulator layout, the focal length can vary by around 500 μm and the grazing angle can vary by around 2 mrad.

From an optics-based viewpoint, the ultracompact mirrors suppress coma aberration, maintaining the diffraction-limited focus even with a relatively large rotation error of 50 μrad in terms of the grazing angle (to be published at Optics Express). This means that the mirror design parameters are not sensitive to the mirror figure.

Regarding the single-aperture approach versus the sMSI approach, we trust the former more. Our sub-meter-radius mirrors have a much stronger curvature than that produced by the systematic errors of interferometers. The mirror radius and mirror figure errors obtained using the single-aperture approach were compared with those evaluated using a coordinate-measuring machine and a traditional MSI approach. The results were in good agreement. However, the results for the sMSI approach slightly deviated from the results for the traditional MSI approach depending on the precision of the translation stages. For the VFM, we used the single-aperture approach as the mirror could fit into the field of view of the interferometer. As the HFM was larger than the field of view, we used the sMSI approach to measure the HFM figures.

C1.10	Lines 739-743. This paragraph is a surprise. It should not be placed in the mirror fabrication section.
-------	---

This paragraph was meant to give the background for the mirror design, which we considered to be related to the mirror fabrication. As this paragraph is also related to the manipulator design, we have moved it to the “SHIMA system” section.

C1.10

Lines 733-743.
 With our efficient metrology approaches, ultracompact mirrors can be measured with a single frame or no more than 10 stitched frames. This evaluation is less error-prone than existing methods.
The incident path length is different between the VFM and the HFM. This configuration stems from the motion reliability required for the secondary slits, which can be achieved using independent vertical and horizontal slits at the beamline. The slits are located downstream of the gratings at beamline BL25SU-A.

Lines 825-841.
 The ucKB mirror was mounted on the SHIMA manipulator with 15 piezo-driven stages for the VFM, HFM, sample, and entrance slits. These four modules were installed within the manipulator dimensions of 138 mm × 133 mm × 250 mm [54]. (...) All detectors and the SHIMA manipulator were under vacuum in the SHIMA chamber. The SHIMA system can synchronize all components.

The incident path length is different between the VFM and the HFM. This configuration stems from the motion reliability required for the secondary slits, which can be achieved using independent vertical and horizontal slits at the beamline. The slits are located downstream of the gratings at beamline Sample specifications BL25SU-A (see Fig. 1a).

C1.11	Section 4.2. It is better to have a sketch of the beamline here.
-------	--

Although a beamline sketch would be helpful in this section, it would repeat some information shown in Fig. 2a. Instead of adding a beamline sketch, we added some more details to Fig. 2a. We also added text to Section 4.2 (“SHIMA system”) to guide the readers to Fig. 2a.

Fig. 2 Experimental configuration using ucKB mirror composed of VFM and HFM. a Layout of soft-X-ray microscope. The transmitted X-rays and diffraction patterns are acquired with a charge-coupled device (CCD) X-ray camera. LEXRF microscopy simultaneously utilizes a silicon drift detector (SDD) and a photo diode (PD) to capture X-ray fluorescence and absorption, respectively. Designed elliptic figure profiles and sub-meter ROCs of b VFM and c HFM. With the positional origin at the mirror center, the positivex-direction is toward the downstream side of the mirror.

C1.11a

Fig. 1 Experimental configuration using ultracompact Kirkpatrick-Baez (ucKB) mirror composed of vertically focusing mirror (VFM) and horizontally focusing mirror (HFM). **a** Layout of soft-X-ray microscope. The transmitted X-rays and diffraction patterns are acquired with a charge-coupled device (CCD) X-ray camera. Low-energy X-ray fluorescence microscopy simultaneously utilizes a silicon drift detector (SDD) and a photo diode (PD) to capture X-ray fluorescence and absorption, respectively. Designed elliptic figure profiles and sub-meter radii of curvature (ROCs) of **b** VFM and **c** HFM. With the positional origin at the mirror center, the positive x-direction is toward the downstream side of the mirror.

Lines 746-757

The ucKB mirror was mounted on the SHIMA manipulator with 15 piezo-driven stages for the VFM, HFM, sample, and entrance slits. These four modules were installed within the manipulator dimensions of 138 mm × 133 mm × 250 mm [54]. (...) All detectors and the SHIMA manipulator were under vacuum in the SHIMA chamber. The SHIMA system can synchronize all components.

Lines 825-841

The ucKB mirror was mounted on the SHIMA manipulator with 15 piezo-driven stages for the VFM, HFM, sample, and entrance slits. These four modules were installed within the manipulator dimensions of 138 mm × 133 mm × 250 mm [54]. (...) All detectors and the SHIMA manipulator were under vacuum in the SHIMA chamber. The SHIMA system can synchronize all components.

C1.11b

The incident path length is different between the VFM and the HFM. This configuration stems from the motion reliability required for the secondary slits, which can be achieved using independent vertical and horizontal slits at the beamline. The slits are located downstream of the gratings at beamline BL25SU-A (see Fig. 1a).

C1.12	Line 820. How the ptychography reconstructed probe function be converted to the figure error?
-------	---

X-ray ptychography can reconstruct the wavefront of the focused X-ray beam, which reflects figure errors on a mirror surface (see Ref. 19: Takeo 2020b, Ref. 60: Seiboth 2017, Ref. 64: Shimamura 2023). The ePIE algorithm was employed for iterative calculations using the obtained images (see Ref. 63: Maiden 2009}. The reconstructed probe function was backpropagated to the downstream side of the mirror. Defocus causes spherical aberration of the wavefront at this position. Because ptychography reconstructs this aberration for some photon energy conditions, the wavefront was first regressed to obtain the best square polynomial. The square-removed wavefront was then converted to the figure error, which is comparable to that obtained using microscopic interferometry.

The above information is given in Section 2 in the Supplementary Information. We added a reference to this section and the above references.

C1.12a	Lines 819-820. The probe function was first reconstructed and then converted to figure errors. Lines 904-907. Ptychography can reconstruct the wavefront of the focused X-ray beam, which reflects figure errors on a mirror surface [19, 57, 61]. The probe function was first reconstructed and propagated to the downstream side of the mirrors. The propagated wavefront was then converted to figure errors.
C1.12b	Lines 831-833. The ptychography results reported in the present study are well grounded in the figure error evaluated using the interferometer. Lines 922-925 The ptychography results reported in the present study are well grounded in the figure error evaluated using the interferometer. See Supplementary Section 2 for detailed figure error reconstruction.

Response to comments from Reviewer 2

The manuscript presents a the design, manufacture, and use of the ultracompact Kirkpatrick-Baez (ucKB) mirror that can focus soft x-ray light to spots of 20 nm by 20 nm over a large photon energy range. This result would be very useful in the fields of optics and for future experiments at X-ray sources, like FELs and synchrotrons, and the results shown here from the experiment are convincing in the new mirror's use.

We are grateful for the recognition of the value of our work.

*Though the subject of the paper is both interesting and relevant, the approach and writing of the paper is hard to follow, meandering, and often does not give the reader all the information one needs.
(...)*

If these points are addressed, I think the paper is worth publishing in Nature Communications.

We appreciate these comments and the detailed suggestions given below. We have updated our manuscript to address the concerns. Please let us know if our responses are satisfactory.

General comments

C2.1	The overall quality of the sentence structure in the mirror manufacture and theory section of the paper could be better. Lots of sentences are set conditionally without reason. A good example is in line 105, where the authors say that novel strategies 'might' have to be developed. Considering that the paper is presenting a novel development, this 'might' is pointless. One should say 'have to be developed' or 'have been developed'.
------	--

As suggested, we have rephrased some expressions in the following sentences. The mirror fabrication and theory sections now use more direct language.

A2.1a	Lines 103-105. To realize ideal nanofocusing over the entire soft-X-ray range, novel strategies might have to be adopted in addition to conventional straightforward development. Lines 105-107 To realize ideal nanofocusing over the entire soft-X-ray range, novel strategies have to be adopted in addition to conventional straightforward development.
A2.1b	Lines 107-110. With the large grazing angle allowed in the soft-X-ray region, millimeter-scale mirrors would moderately accept X-rays and achieve focusing throughput superior or comparable to that for diffractive focusing devices [3, 8]. Lines 109-112. With the large grazing angle allowed in the soft-X-ray region, millimeter-scale mirrors moderately accept X-rays and achieve focusing throughput superior or comparable to that for diffractive focusing devices [3, 8].
A2.1c	Lines 118-119. Millimeter-scale focal lengths would enhance focusing robustness as X-rays scattered by mirror slope errors reach the focal plane before widely spreading (see Fig. 4). Lines 118-119. Millimeter-scale focal lengths enhance focusing robustness as X-rays reach the focal plane before widely spreading due to mirror figure errors (see Fig. 4).

Lines 118-119.
Such ultracompact mirrors ~~could~~ be simple alternatives for ideal nanofocusing.

A2.1d Lines 120-121.
Such ultracompact mirrors can be simple alternatives for ideal nanofocusing.

Lines 122-123.
A robust focusing strategy, reinforced by new technology, ~~may~~ enable the realization of ideal achromatic soft-X-ray nanoprobe.

A2.1e Lines 124-125.
A robust focusing strategy, reinforced by new technology, should enable the realization of ideal achromatic soft-X-ray nanoprobe.

Lines 130-133.
An ultracompact KB (ucKB) mirror with component focal lengths of 2 and 8 mm is designed based on a short-focal-length strategy. ~~The aim is a sub-50-nm nanoprobe with a photon energy of more than 1 keV, which would give access to mesoscopic scales below 100 nm, where magnetic skyrmions [10] and subcellular biological behavior [25] between bulk and nanoscale properties emerge.~~

A2.1f Lines 133-136.
An ultracompact KB (ucKB) mirror with component focal lengths of 2 and 8 mm is designed based on a short-focal-length strategy and then fabricated using our advanced methods. The ucKB mirror achieves a sub-50-nm nanoprobe with a photon energy of more than 1 keV, which gives access to mesoscopic scales below 100 nm, where magnetic skyrmions [10] and subcellular biological behavior [25] between bulk and nanoscale properties emerge.

Lines 146-148.
~~Wave optics can also be used to deduce the diffraction effects triggered by practical figure errors.~~

A2.1g Lines 149-151.
The focus spot diffracted by figure errors is deduced using wave optics::

Introduction

C2.2	The introduction of the paper gives the impression that the paper is theoretical, about how to build the ucKB. It is only from section 2.2 onward that the reader realizes that the mirror is manufactured and an experiment with it is done. The introduction should more clearly state that the authors have design the ucKB, manufactured it, and did an experiment with it that showed new results.
------	---

We have modified the last paragraph in the Introduction section to clearly state that the mirror design and manufacture, an experiment using the ucKB, and the results obtained are discussed in the manuscript.

Here, besides developing fabrication techniques for highly curved mirrors, we examine remarkably compact mirrors and short focal lengths for nanofocusing soft X-rays. The proposed mirror system employs a sequentially crossed arrangement referred to as the Kirkpatrick-Baez (KB) geometry [24] (see Fig. 2), which simplifies the shape of the individual mirrors. An ultracompact KB (ucKB) mirror with component focal lengths of 2 and 8 mm is designed based on a short-focal-length strategy. ~~The aim is a sub-50-nm nanoprobe~~

with a photon energy of more than 1 keV, which ~~would give~~ access to mesoscopic scales below 100 nm, where magnetic skyrmions [10] and subcellular biological behavior [25] between bulk and nanoscale properties emerge. We apply ~~an~~ ucKB mirror to enhance soft-X-ray fluorescence microscopy, namely the low-energy X-ray fluorescence (LEXRF) technique [26], ~~for~~ multicolor soft-X-ray ~~nanoanalyses~~.

A2.2 Here, besides developing fabrication techniques for highly curved mirrors, we examine remarkably compact mirrors and short focal lengths for nanofocusing soft X-rays. The proposed mirror system employs a sequentially crossed arrangement referred to as the Kirkpatrick-Baez (KB) geometry [24] (see Fig. 2), which simplifies the shape of the individual mirrors. An ultracompact KB (ucKB) mirror with component focal lengths of 2 and 8 mm is designed based on a short-focal-length strategy and then fabricated using our advanced methods. The ucKB mirror achieves a sub-50-nm nanoprobe with a photon energy of more than 1 keV, which gives access to mesoscopic scales below 100 nm, where magnetic skyrmions [10] and subcellular biological behavior [25] between bulk and nanoscale properties emerge. We apply the ucKB mirror to enhance soft-X-ray fluorescence microscopy, namely the low-energy X-ray fluorescence (LEXRF) technique [26]. The results of observations of fixed biological specimens demonstrate the feasibility of a multimodal and multicolor soft-X-ray nanoanalysis.

Results

C2.3	The results section has equations and modeling. This is very odd for a results section, since I would expect the theoretical consideration of the mirror to be in the 'setup' or 'experiment section'. To that point, the paper has neither of these sections, which is not something one sees very often. I would strongly recommend the authors re-consider the order and the headings of the sections of the manuscript.
------	---

We agree that a separate section for theoretical considerations, such as a “Setup” or “Experiment” section, would be useful right before the Results section. However, the guidelines for *Nature Communications* strictly limit the sections to Introduction, Results, Discussion, Conclusion, and Methods. Only the Results and Methods sections are allowed to have subsections. As a short-focal-length strategy is for the first time applied to X-ray mirrors, we believe that the theoretical considerations have some novelty and belong in the Results section. Our original manuscript thus showed equations and modeling in the Results section with the heading “Ultrashort focal length and mirror length”. This subsection led readers to the following design subsection (“Design and fabrication of ultracompact focusing mirrors”).

In the revised manuscript, we have moved most of the mathematical considerations into a new subsection in the Methods section, entitled “Ultrashort focal length and mirror length”. The last mathematical consideration, namely Eq. (1) [Eq. (3) in the original manuscript], has been added to the subsection “Design of ultracompact X-ray mirrors” in the Results section. This equation is necessary for the Discussion section and it would thus be inconvenient for readers to have it in the Methods section.

If this modification is insufficient, we can discuss this issue with the Editor of *Nature Communications*.

Lines 142-207.

Figure 1 compares short and long mirrors with figure errors for a given spatial wavelength. Figure errors cause an uneven slope distribution, which scatters rays in proportion to the focal length. From a geometrical viewpoint, short focal lengths mitigate the effects of mirror defects on ray deflection.

Wave optics can also be used to deduce the diffraction effects triggered by practical figure errors. Figure 1a shows a wavefield traveling along the positive z direction from the aperture on the x axis to the focal area across the u axis. An elliptical X ray mirror crops the incident wavefield into an aperture shape and the surface imperfection disturbs the wavefront. The grazing angle of X ray mirrors is approximately constant around θ_g . As illustrated in the inset in Fig. 1a, a periodic figure error with amplitude h_0 and spatial wavelength d_m is thus assumed to be replicated to the aperture wavefront with height error $h_a = 2h_0 \sin \theta_g$ and spatial wavelength $d_a = d_m \sin \theta_g$ [27]. The wavefront at the aperture U_a modulated by the wavefront error P_m can be modeled as

$$U_a^{\text{error}}(x) = P_m(x)U_a^{\text{ideal}}(x) \\ \approx \exp\left[j\frac{2\pi}{\lambda}h_a \cos\left(\frac{2\pi}{d_a}x\right)\right]U_a(x)\text{rect}\left(\frac{x}{a}\right)$$

where $2a$ is the aperture width, λ is the wavelength of the monochromatic light, U_i is the incident wavefield, which is normally uniform, and $\text{rect}(x/a)$ returns non zero values under $|x| \leq a$.

The wavefield circularly transformed by the focusing device follows the Fraunhofer diffraction formula [28]. Provided that h_a is significantly small and the NA is constant at α ($\arctan(a/r) \approx a/r \approx \alpha$), the focused wavefield is expressed as

$$U_f(u) = A(u; r) r \alpha \left[\mathcal{F}[P_m U_i] \left(\frac{u}{\lambda r} \right) * \text{sinc} \left(\frac{\alpha u}{\lambda} \right) \right] \\ \approx A(u; r) r \alpha \left\{ (1 - \phi^2) \text{sinc} \left[\frac{\alpha u}{\lambda} \right] \right. \\ \left. - \frac{\phi^2}{2} \left(\text{sinc} \left[\frac{\alpha}{\lambda} \left(u - \frac{2\lambda r}{d_m} \right) \right] + \text{sinc} \left[\frac{\alpha}{\lambda} \left(u + \frac{2\lambda r}{d_m} \right) \right] \right) + j\phi \left(\text{sinc} \left[\frac{\alpha}{\lambda} \left(u - \frac{\lambda r}{d_m} \right) \right] + \text{sinc} \left[\frac{\alpha}{\lambda} \left(u + \frac{\lambda r}{d_m} \right) \right] \right) \right\} \\ A(u; r) = \frac{1}{j\lambda r} \exp \left[\frac{j\pi u^2}{\lambda r} \right]$$

where U_f is the focus wavefield, \mathcal{F} is the Fourier transform operator, the symbol $*$ denotes the convolution operation, $\phi = \pi h_a / \lambda$, and r is the distance between the focus and the downstream mirror end.

Five normalized sine functions underlie U_f . The first term is independent of r and centered at $u = 0$ (resulting from the ideal focus); it shows that smooth mirrors with the same NA form an identical focus spot regardless of focal length. The other terms are centered either at $u = \pm \lambda r / d_m$ or $\pm 2\lambda r / d_m$, whose positions diverge with r . They express extra focus spots diffracted by surface imperfections. The gap between the extra focus spots Δu is

$$\Delta u = \frac{\lambda r}{d_m} = \frac{\lambda r}{d_m \sin \theta_0} \approx \frac{\lambda r}{d_m \theta_0}$$

In principle, foci close to the devices are beneficial for realistic nanofocusing systems. Figure errors of $d_m = 500 \mu\text{m}$ have been observed during mirror production. The soft X-ray region spans about $\lambda = 1.24 \text{ nm}$ (1 keV photon energy). A reflectivity of 50% restricts θ_0 to approximately 25 mrad or less [29]. Access to the mesoscopic scale requires $\Delta u = 100 \text{ nm}$. Substituting these parameters into Eq. (3) yields $r \approx 1 \text{ mm}$.

An r value of 1 mm goes against the traditional design philosophy for hard X-ray mirrors. Diffractive nanofocusing devices often employ an r value of tens to hundreds of micrometers [3]. Even though post-production simulation can be used to numerically evaluate how figure errors affect focusing performance [27], the short focal length strategy can lift fabrication restrictions in advance, as shown in Fig. 1b. A detailed mathematical derivation is provided in Supplementary Section 1.

Lines 146-175.

Figure errors cause an uneven slope distribution, which scatters rays in proportion to the focal length. From a geometrical viewpoint, short focal lengths mitigate the effects of mirror defects on ray deflection (see Figs. 7a and b). The focus spot diffracted by figure errors is deduced using wave optics:

$$\Delta u \approx \frac{\lambda r}{d_m \theta_0}$$

where Δu is the gap between the diffracted focus spots, λ is the wavelength of monochromatic X-rays, r is the distance between the focus and the downstream mirror end, d_m is the spatial wavelength of a periodic figure error, and θ_0 is the grazing angle of X-ray mirrors (see Methods). In principle, foci close to the devices are beneficial for realistic nanofocusing systems. Equation (1) with empirical values shows that soft-X-ray nanofocusing mirrors promising for the mesoscopic scales require $r = 1 \text{ mm}$ (see Methods). Diffractive nanofocusing devices often employ an r value of tens to hundreds of micrometers [3]. However, an r value of 1 mm goes against the traditional design philosophy for hard-X-ray mirrors.

A2.3a

As illustrated in Fig. 1a, the KB geometry simplifies a doubly curved focusing mirror to a pair of elliptic-cylindrical surfaces, namely the vertically focusing mirror (VFM) and horizontally focusing mirror (HFM). However, one mirror comes between the other mirror and the focus, and the downstream mirror prevents a short upstream focal length. To achieve an extremely short focal length for both mirrors of the ucKB mirror, the downstream mirror length was reduced to 2 mm. Table 1 lists the detailed optical design parameters for the VFM and HFM. The r value for the VFM is 1 mm. The ucKB mirror was designed to concentrate 1-keV soft X-rays into a sub-50-nm focus spot, which is sufficient for mesoscopic-scale analyses.

The large NA and demagnification factor require a sub-meter-radius design, as shown in Figs. 1b and c. Such strongly curved mirrors are the most critical challenge in terms of advanced production and metrology.

(New sections)

Lines 658-784.
Methods

Ultrashort focal length and mirror length

Figure 7 compares short and long mirrors with figure errors for a given spatial wavelength. Figure 7a shows a wavefield traveling along the positive z-direction from the aperture on the x-axis to the focus area across the u-axis. An elliptical X-ray mirror crops the incident wavefield into an aperture shape and the surface imperfection disturbs the wavefront. The grazing angle of X-ray mirrors is approximately constant around \$\theta_0\$. As illustrated in the inset in Fig. 7a, a periodic figure error with amplitude \$h_0\$ and spatial wavelength \$d_m\$ is thus assumed to be replicated to the aperture wavefront with height error \$h_a = 2h_0 \sin \theta_0\$ and spatial wavelength \$d_a = d_m \sin \theta_0\$ [52]. The wavefront at the aperture \$U_a\$ modulated by the wavefront error \$P_m\$ can be modeled as

$$\begin{aligned} U_a^{\text{error}}(x) &= P_m(x)U_a^{\text{ideal}}(x) \\ &\approx \exp\left[j\frac{2\pi}{\lambda}h_a \cos\left(\frac{2\pi}{d_a}x\right)\right]U_i(x)\text{rect}\left(\frac{x}{a}\right), \end{aligned}$$

where $2a$ is the aperture width, λ is the wavelength of the monochromatic light, U_i is the incident wavefield, which is normally uniform, and $\text{rect}(x/a)$ returns non-zero values under $|x| \leq a$.

The wavefield circularly transformed by the focusing device follows the Fraunhofer diffraction formula [28]. Provided that h_0 is significantly small and the NA is constant at α ($\arctan(a/r) \approx a/r \approx \alpha$), the focused wavefield is expressed as

$$\left\{ \begin{aligned} U_f(u) &= A(u;r)r\alpha \left[\mathcal{F}\{P_m U_i\} \left(\frac{u}{\lambda r} \right) * \text{sinc} \left(\frac{\alpha u}{\lambda} \right) \right] \\ &\approx A(u;r)r\alpha \left\{ (1 - \phi^2) \text{sinc} \left[\frac{\alpha u}{\lambda} \right] \right. \\ &\quad \left. + \frac{\phi^2}{2} \left(\text{sinc} \left[\frac{\alpha}{\lambda} \left(u - \frac{2\lambda r}{d_a} \right) \right] + \text{sinc} \left[\frac{\alpha}{\lambda} \left(u + \frac{2\lambda r}{d_a} \right) \right] \right) \right\} + j\phi \left(\text{sinc} \left[\frac{\alpha}{\lambda} \left(u - \frac{\lambda r}{d_a} \right) \right] + \text{sinc} \left[\frac{\alpha}{\lambda} \left(u + \frac{\lambda r}{d_a} \right) \right] \right) \\ A(u;r) &= \frac{1}{j\lambda r} \exp \left[\frac{j\pi u^2}{\lambda r} \right], \end{aligned} \right.$$

where U_f is the focus wavefield, \mathcal{F} is the Fourier transform operator, the symbol $*$ denotes the convolution operation, $\phi = \pi h_a / \lambda$, and r is the distance between the focus and the downstream mirror end.

Five normalized sinc functions underlie U_f . The first term is independent of r and centered at $u = 0$ (resulting from the ideal focus); it shows that smooth mirrors with the same NA form an identical focus spot regardless of focal length. The other terms are centered either at $u = \pm \lambda r / d_a$ or $\pm 2\lambda r / d_a$, whose positions diverge with r . They express extra focus spots diffracted by surface imperfections. The gap between the extra focus spots Δu is

$$\Delta u = \frac{\lambda r}{d_a} = \frac{\lambda r}{d_m \sin \theta_0} \approx \frac{\lambda r}{d_m \theta_0},$$

as described in Eq. (1). Figure errors of $d_m = 500 \mu\text{m}$ have been observed during mirror production. The soft-X-ray region spans about $\lambda = 1.24 \text{ nm}$ (1-keV photon energy). A reflectivity of 50% restricts θ_0 to approximately 25 mrad or less [54]. Access to the mesoscopic scale requires $\Delta u = 100 \text{ nm}$. Substituting these parameters into Eq. (5) yields $r \approx 1 \text{ mm}$.

The advantage of a short focal length was examined using an in-house wave propagation code based on the Fresnel-Kirchhoff diffraction formula [55]. Large figure errors were randomly generated under the assumption that their spatial frequency spans a range of 500 to 2000 μm^{-1} . Such errors can remain after mirror fabrication. To simulate fabrication-constrained mirrors, the amplitudes of the figure errors were increased beyond the range determined by Rayleigh's quarter wavelength rule, as shown in Fig. 7c. Then, the focusing profiles generated by mirrors with different focal lengths were calculated with the figure errors fed into the simulation. The mirror length was adjusted to achieve the same NA (0.0149) with different focal lengths. Figure 7d shows that short-focal-length mirrors can generate a clear main peak, especially with large figure errors.

Even though post-production simulation can be used to numerically evaluate how figure errors affect focusing performance [52, 55], the short-focal-length strategy can lift fabrication restrictions in advance, as shown in Fig. 7b-d. A detailed mathematical derivation is provided in Supplementary Section 1.

C2.4	The performance in section 2.3 and the figure 4 do not answer some important questions. For example, the authors, in the introduction, mention that one of the major issues with current solutions for soft X-ray focusing leads to energy scans not being easily doable due to the changing position of the focus with photon energy. However, there is no discussion whether the focus stays stable with the ucKB, and no data showing the focus staying on point over different energies in figure 4. They should have a graph similar to 4a for the X, Y, and Z position of the focus as a function of photon energy. The supplementary information provides good information, but an overview graph should be put in the main article to show how the mirror is superior to the previous solutions in every way mentioned in the paper.
------	---

X-ray mirrors are based on reflection, which is an achromatic phenomenon in principle, to focus X-rays. We thought that the use of X-ray mirrors was sufficient to verify achromatic soft-X-ray focusing. The focus size of achromatic soft-X-ray nanobeams is inversely proportional to photon energy, as shown in Fig. 4a. If our X-ray mirrors had not been achromatic, the focus size would not follow such a curve, as is the case for diffractive focusing devices. A Foucault knife-edge test did not show any signs of defocus. The STXM images of Au nanoparticles were taken at 2-keV photon energy without adjusting the focus position after ptychography was performed at 1-keV photon energy. The ptychographically reconstructed wavefronts did not contain a significant square component, which stems from defocus. The bicolor nanoprobe were formed by mixing 1- and 2-keV photon energies and achieved a 100-nm spatial resolution. These results show that the X-ray mirrors produce achromatic focus spots and do not change the focus position over different energies.

In the revised manuscript, the in-plane shifts (X and Y) of the focus position and the amount of defocus (Z) are shown in Fig. 4. The in-plane shifts were evaluated using ptychography and knife-edge scanning. The amount of defocus was evaluated by backpropagating ptychographically reconstructed wavefields to the downstream of the X-ray mirror and calculating the difference in the radius of curvature of the wavefronts. The results are within the error ranges. The change in the focus position is very small compared to the chromatic aberration produced by other focusing devices.

Fig. 4 Achromatic soft-X-ray nanoprobe formed by ~~ucKB~~ mirror. *a* FWHM focus size versus X-ray photon energy. The knife-edge scanning and ptychography methods both reveal that the increased photon energy condensed soft X-rays into a smaller focus spot. *b* Ptychographically reconstructed focusing profiles with an FWHM spot size of 20.4 nm × 40.7 nm at 2-keV photon energy. *c* SEM and *d* ptychography images of Au nanoparticles. The ptychography image was captured at 1-keV photon energy with a 10.3-nm pixel size. The Au particles were mounted on a Si₃N₄ supporting film. The weak electric conductivity of this film causes charge-up effects, which manifest as a streak in the SEM image. Scale bars in *c* and *d*: 1 μm. *e* SEM

micrograph and 2-keV STXM image of single Au nanoparticle. The bright area differs in the images as SEM detects electron scattering from the surface and STXM measures the transmittance. The dashed lines highlight the outline of the nanoparticle. The sidelobe in **b** causes satellite signals (see arrows). Scale bar: 100 nm. The X-ray images in **d** and **e** were captured at the identical focus position. The positive coordinates follow the directions in Fig. 2. **f** Comparison of figure errors converted from ptychographic probe functions and observed using interferometry. The specimens given in parentheses are described in Methods. With the positional origin at the mirror center, the positive x-direction is toward the downstream side of the mirror.

A2.4a

Fig. 3 Achromatic soft-X-ray nanoprobe formed by ultracompact Kirkpatrick-Baez mirror. **a** Full width at half maximum (FWHM) focus size and focus position versus X-ray photon energy. The error ranges for the in-plane shift and the defocus amount were calculated based on \pm FWHM focus size and Rayleigh range, respectively. The knife-edge scanning and ptychography methods both reveal that the increased photon energy condensed soft X-rays into a smaller focus spot. **b** Ptychographically reconstructed focusing profiles with an FWHM spot size of $20.4 \text{ nm} \times 40.7 \text{ nm}$ at 2-keV photon energy. **c** Scanning electron microscopy (SEM) and **d** ptychography images of Au nanoparticles. The ptychography image was captured at 1-keV photon energy with a 10.3-nm pixel size. The Au particles were mounted on a silicon-nitride supporting film. The weak electric conductivity of this film causes charge-up effects, which manifest as a streak in the SEM image. Scale bars in **c** and **d**: 1 μm . **e** SEM micrograph and 2-keV scanning transmission X-ray microscopy (STXM) image of single Au nanoparticle. The bright area differs in the images as SEM detects electron scattering from the surface and STXM measures the transmittance. The dashed lines highlight the outline of the nanoparticle. The sidelobe in **b** causes satellite signals (see arrows). Scale bar: 100 nm. The X-ray images in **d** and **e** were captured at the identical focus position. The positive coordinates follow the directions in Fig. 1. **f** Comparison of figure errors converted from ptychographic probe functions and observed using interferometry. The specimens given in parentheses are described in Methods. With the positional origin at the mirror center, the positive x-direction is toward the downstream side of the mirror.

We have also added a description of the evaluation of the focus position to the Supplementary Information.

(New sections)

Supplementary information
3.3 Focus position

A2.4b

A Foucault knife-edge test showed that the focus spot remained within the Rayleigh range. The knife edge was therefore scanned at the same position along with the X-ray path. The in-plane shifts were calculated from the center of the Gaussian-fit profiles. The small shifts of the focus position shown in Fig. 3a can

stem from the transparency of the knife edge to high-energy X-rays. The secondary slits (S2 and SCa) and the monochromator were also moved to change photon energy; such movement of the upstream optics may have contributed to the drift in the focus spot and caused the mirror surface illuminated by X-rays to be shifted in the tangential direction.

The amount of defocus was evaluated by backpropagating ptychographically reconstructed wavefields to the downstream of the X-ray mirror and calculating the difference in the radius of curvature of the wavefields. Using the coordinate system shown in Fig. 7a, the circular wavefront that is ideally formed with a smooth mirror is expressed as a part of a circle:

$$x^2 + (z - r)^2 = r^2.$$

The wavefront is located at the downstream side of the mirror, which usually has a small spatial acceptance compared to the working distance (i.e. $x/r \ll 1$).

$$\begin{aligned} z &= r + r \sqrt{1 - \frac{x^2}{r^2}} \\ &\approx 2r - \frac{x^2}{2r} \text{ or } \frac{x^2}{2r} \\ \therefore z &\approx \frac{x^2}{2r}. \end{aligned}$$

The defocus amount Δr is the same as the difference in the radius of curvature of the wavefront. If the coefficient of the square term for the reconstructed wavefront is a ,

$$\begin{aligned} z &= \frac{x^2}{2(r + \Delta r)} - \frac{x^2}{2r} \\ &= -\frac{\Delta r}{2(r + \Delta r)r} x^2 \\ &= ax^2. \end{aligned}$$

Therefore,

$$\Delta r = -\frac{2r^2 a}{2ra + 1}.$$

Reviewers' Comments:

Reviewer #1:

Remarks to the Author:

The authors have responded to my previous comments. I do not have further comments.

Reviewer #2:

Remarks to the Author:

The changes to the article have answered the initial issues that were presented about the article, and I find its structure, both grammatically and in its layout, to be significantly better and easier to read and understand. The clarification about why the headings were chosen as they were also helps to clear up a few issues.

In my opinion, the article, as it is, is suitable for publication.